



# Flood estimation for ungauged catchments in the Philippines

Trevor B. Hoey[1], Pamela Louise M. Tolentino[2,3], Esmael L. Guardian[3], John Edward G. Perez[3,4], Richard D. Williams[2], Richard J. Boothroyd[5], Carlos Primo C. David[3] and Enrico C. Paringit[6,7]

[1] Department of Civil and Environmental Engineering, Brunel University London, London, UB8 3PH, United Kingdom
[2] School of Geographical and Earth Sciences, University of Glasgow, Glasgow, G12 8QQ, United Kingdom
[3] National Institute of Geological Sciences, University of the Philippines Diliman, Philippines
[4] University of Vienna, Vienna, Austria
[5] Department of Geography and Planning, University of Liverpool, Liverpool, L69 7ZT, United Kingdom
[6] Department of Geodetic Engineering, University of the Philippines Diliman, Philippines
[7] Department of Science and Technology - Philippine Council for Industry, Energy and Emerging Technology Research and Development

*Correspondence to*: Pamela Louise M. Tolentino (Pammie.Tolentino@glasgow.ac.uk)

## 1 Abstract

Flood magnitude and frequency estimation are essential for the design of structural and nature-based flood risk management interventions and water resources planning. However, the global geography of hydrological observations is uneven; in many regions, such as the Philippines, data are spatially and/or temporary sparse, limiting the choice of statistical methods for flood estimation. We evaluate the potential of pooling short historical data series for ungauged catchment flood estimation. Daily mean river discharge data were collected from 842 sites, with data spanning from 1908 to 2018. Of these, 513 candidate sites met criteria to estimate a reliable annual maximum flood. Using the index flood approach, a range of controls were assessed at national and regional scales using land cover and rainfall datasets, and GIS-derived catchment characteristics. Multivariate analysis for predictive equations for 2 to 100 year recurrence interval floods based on catchment area only have $R^2 \leq 0.59$. Additionally, adding a rainfall variable, the median annual maximum 1-day rainfall, increases $R^2$ to between 0.56 for $Q_{100}$ and 0.66 for $Q_2$. Very few other variables were significant when added to multiple regression equations. Although the Philippines exhibits regional climate variability, there is limited spatial structure in predictive equation residuals and region-specific predictive equations do not perform significantly better than national equations. Relatively low $R^2$ values are typical of studies from tropical regions. The predictive equations are suitable for use as design equations for the Philippines but uncertainties must be assessed. Our approach demonstrates how combining individually short historical records, after careful screening and exclusion of erroneous data, generates large data sets that can produce consistent results. Extension of continuous flood records is required to reduce uncertainties but national-scale consistency suggests that extrapolation from a small number of carefully selected catchments could provide nationally reliable predictive equations with reduced uncertainties.

## 2 Introduction and rationale

The impact of river flooding across Southeast Asia is severe on a global scale, whether measured in terms of inundated area, the number of people affected or fatalities (Ziegler et al., 2020). Understanding the hazard and designing mitigation or adaptation strategies relies on estimating flood magnitude and frequency, which is achieved through empirical analyses of available data and, for forecasting, the results of climate and hydrological



models. The resulting equations to estimate flows of specified recurrence are used for a wide range of purposes including insurance loss estimation (Lyubchich et al., 2018), aquatic biodiversity assessment (Parasiewicz et al., 2019), engineering design and water resource planning.

A wide range of statistical methods have been applied to flood frequency estimation (see Asquith et al., 2017 for a recent listing). The index flood approach uses the median or mean annual maximum flood, or equivalently a flood of specified recurrence interval, and relates this to catchment properties to develop regional predictive equations (eg Dalrymple, 1960; Kjeldsen and Jones, 2006; Stedinger and Lu, 1995). In data-rich settings, such approaches can be complex, as illustrated by the United Kingdom (UK) Flood Estimation Handbook (FEH).

Kjeldsen et al. (2008; Table 4.1) show how successive iterations of predictive equations for the UK have added variables and statistical complexity. However, catchment area and annual precipitation remain the most significant predictors even in this case (Meigh et al., 1997). Although the index flood method is reliable and can yield high $R^2$ values, adding non-linear effects and spatially-dependent interactions have been proposed as potential sources of further improvement (Muhammad and Lu, 2020).

In many countries and regions data may be sparse in space and/or time (Mamun et al., 2011), limiting the choice of statistical methods for flood frequency estimation and strongly influencing the magnitude of associated uncertainties. The lengths of records that are available impacts on the analytical results (Fischer and Schumann, 2022), and uncertainty increases with short data series. This uncertainty can be reduced by extending data series through use of historical or proxy information (Macdonald et al., 2014; Merz and Blöschl, 2008; Reinders and

Muñoz, 2021; Ziegler et al., 2020), by cross-validation against hydrological modelling predictions (Haberlandt and Radtke, 2014), or by pooling information from many sites (Kjeldsen, 2015; Griffiths et al., 2020).

For the Philippines, which exemplifies some of the challenges of using sparse hydrological data, some national-scale analyses of flood magnitude and frequency have been undertaken. Meigh (1995) analysed data, mostly from up to 1980, from 333 sites collected by the BRS (Bureau of Research and Standards). Growth curves and

prediction equations for flood magnitude were presented for different hydrological regions and catchment sizes (Meigh, 1995; Meigh et al., 1997). Liongson (2004) demonstrated a significant relationship between catchment area and mean annual flood ($Q_{MAF}$) for 29 sites in northern Luzon, and analysed the form of growth curves. Regional differences in climate and precipitation patterns are well-documented (Bagtasa, 2017) and projections have been made of climate change impacts on river flow (Tolentino et al., 2016) with some evidence for significant

changes having occurred in recent decades (Meigh, 1995). Calibrating local data with global runoff datasets enables the augmentation of catchment-specific data to a certain extent (Ibarra et al 2021).

Studies of flood magnitude across South-East Asia provide valuable regional context for our Philippines analysis. Loebis (2002) found significant correlations between mean annual flood and catchment area in Indonesia, Laos and Thailand, as did Meigh et al. (1997) for Indonesia, Papua New Guinea and Thailand. Mamun et al. (2011)

provide updated equations for peninsular Malaysia which use catchment area and mean annual rainfall as predictors. In these studies, coefficients of determination ($R^2$) values range from 0.5 to 0.9 tending to be higher in smaller regions and where inter-annual rainfall variability is lower: for example, Meigh et al. (1997) report $R^2$ values of 0.92 for Papua New Guinea and 0.46 for regions 3-8 in the Philippines.



There are few continuous long river flow records available for the Philippines, but many short (3-35 years) records
exist from all regions of the country. This scarcity of data leads to the Philippines being omitted from databases
used for global flow frequency analyses (e.g. Zhao et al., 2021). Pooling of the information from the available
records, taking account of climatic variability across the country, forms the basis of the analysis in this paper. The
approach uses elements of the UK Flood Estimation Handbook methodology, adapted to reflect the nature of the
river flow and other data that are available. The paper aims to evaluate the potential of pooling short data series
to deliver estimates of flood magnitude and frequency that can be applied in ungauged catchments, and hence to
deliver design equations that can be applied across the Philippines.

**3 Data sources**

Daily mean river discharge data were collated from 842 sites (Table 1) reported by three sources. (1)  The "SWS"
data set comes from four volumes of the "Surface Water Supply of the Philippine Islands" (Irrigation Division,
1923-4) contain rating curves and daily flow measurements over the period 1908 – 1922. Water level
measurements were made at constructed weirs and rating curves were computed using discharges obtained by the
velocity-area method. Rating information is supported by detailed information on the measurement site, bank and
bed characteristics and river channel stability. Data from 248 SWS stations across the country (Figure 1) were
used. (2) The second dataset ("BRS") was initially managed by the Bureau of Research Standards, later being
transferred to the Bureau of Design, also under the Department of Public Works and Highways (DPWH). The
BRS data set (Figure 1) is in three parts: BRS_A contains 364 gauging sites with data in the period 1940-1980,
BRS_B has a further 181 with data from 1980 onwards. BRS_C includes 27 of the sites from BRS_A and BRS_B
that are either at identical locations or are sufficiently close (within a few km, without any significant tributaries
in between) to allow for their records to be combined. This produces a maximum record length of 62 years. Some
of these sites had automated water level sensors but most sites had a gauging structure at which manual
observations were made three times per day.  Rating curves were obtained by velocity-area gauging. (3) The
source of the third dataset ("Cagayan") is the "Feasibility Study of the Flood Control Project for the Lower
Cagayan River in the Republic of the Philippines" produced by Nippon Koei Co. and Nikken Consultants Inc. in
collaboration with the DPWH in 2002 (Nippon Koei, 2002). This study only considers the Cagayan watershed,
north Luzon, the largest catchment in the Philippines. Out of 78 gauging stations in the watershed, 48 stations
(Figure 1) were used in this study since some of the stations only reported gauge height data and others have a lot
of gaps. Daily mean water level data were recorded from 1955 to 1991 and converted to discharge using rating
curves (details not reported; Nippon Koei, 2002).

The data were initially filtered to remove sites with very short records (<7 years), inadequate rating between water
level and discharge and those from the SWS data set where the gauging site location could not be reliably
determined.



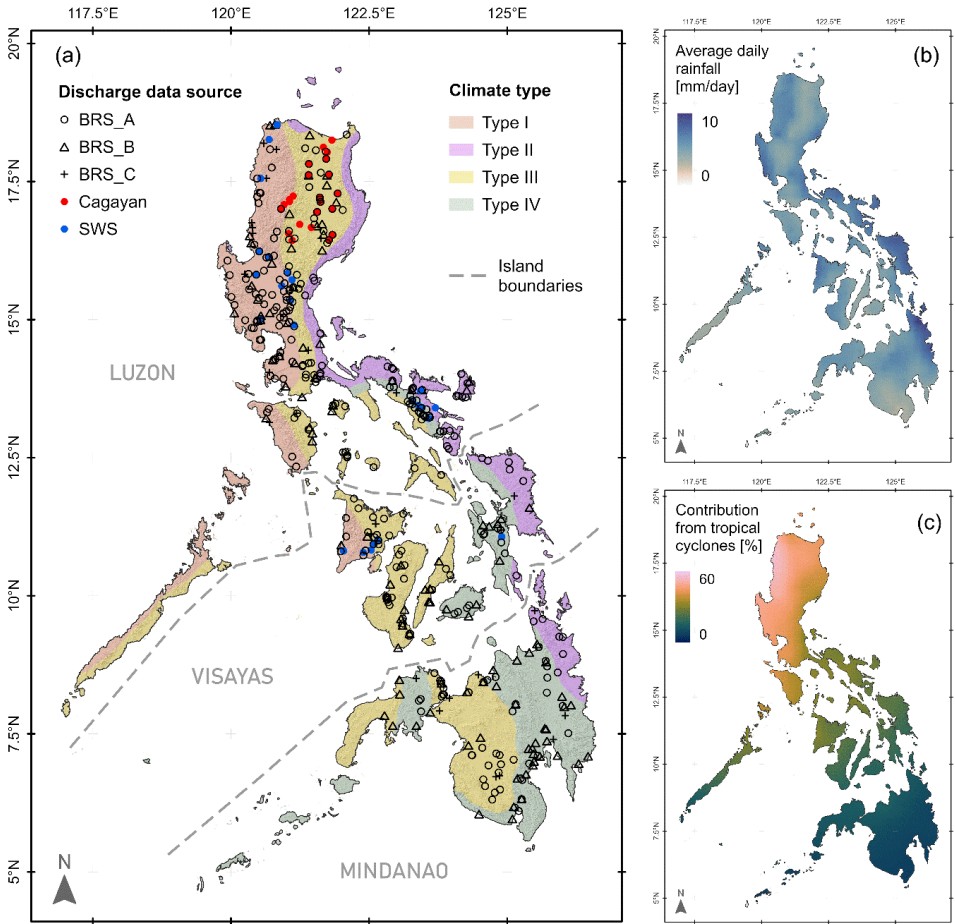

**Figure 1:** (A) Locations of gauging sites from the data sources used in the analysis (n=466; Table 2). Background map (after Tolentino et al., 2016) shows elevation shading overlain by the four climate types that have been identified for the Philippines (Coronas, 1920). (B) Mean daily rainfall (after Bagtasa, 2017). (C) Proportion of annual rainfall generated by tropical cyclones (after Bagtasa, 2017). The climates can be summarised as (Ibarra et al., 2021): type I -distinct wet and dry seasons; type II - no distinct dry season and relatively high rainfall; type III – lower overall rainfall with short dry and wet seasons; and, type IV - reasonably even distribution with lower total rainfall.





**Table 1** Summary of available discharge data sets. Candidate sites are sites retained after removing sites with no
or poor rating, or indeterminate locations. Record length is the number of years for which reliable annual
maximum flow estimates exist, after removal of erroneous data.

| Source | Time period of data | Total number of sites | Number of candidate sites | Number of candidate sites with ≥7 years record | Record Length (years) for sites with ≥7 years data (figures in brackets are for all candidate sites) | | |
|---|---|---|---|---|---|---|---|
| | | | | | Max | Mean | Total |
| SWS | 1908 - 1922 | 248 | 119 | 30 | 10 | 7.7 (5.1) | 230 (604) |
| BRS_A | 1940 - 1980 | 364 | 337 | 310 | 34 | 18.3 (17.1) | 5659 (5771) |
| BRS_B | 1980 - 2018 | 154 | 144 | 115 | 33 | 16.1 (13.9) | 1856 (2003) |
| BRS_C | 1940 - 2018 | 27 | 27 | 27 | 62 | 36.2 (36.2) | 978 (978) |
| Cagayan | 1955 - 1991 | 49 | 46 | 31 | 20 | 11.6 (9.5) | 361 (437) |
| TOTAL | | 842 | 673 | 513 | 62 | 17.7 (14.6) | 9084 (9793) |

## 4 Analysis Methods

### 4.1 Curve fitting for annual daily maximum flows

The maximum flows in each calendar year were extracted from the daily flow data and fitted with three
distributions: (1) Generalised Logistic Distribution (GLO) (Kjeldsen and Jones, 2006; Kjeldsen, 2013); (2)
Weibull; (3) Log-Pearson Type III. The median annual flood (*Qmed*) was used as the index flood, rather than the
mean, to minimise the effect of outliers in the data (Kjeldsen and Jones, 2006), and the parameters of the
distributions were estimated using L-moments (Hosking, 1990; Hosking and Wallis, 1997). L-moments are linear
combinations of probability-weighted moments, and the GLO distribution uses ratios between the first three L-
moments, $l_1$, $l_2$ and $l_3$, to define the L-CV (coefficient of variation) $t_2$ and L-Skewness $t_3$ as:

$$t_2 = l_2/l_1 \qquad t_3 = l_3/l_2 \qquad\qquad (1).$$

The GLO is a three parameter distribution, which has location, scale and shape parameters. The location ($\xi$) is the
median of the distribution. The shape ($\kappa$) and scale ($\beta$) parameters are estimated from the L-moment ratios (Eq.
1), as:

$$\hat{\kappa} = -t_3 \qquad \hat{\beta} = \frac{t_2 \hat{\kappa} sin(\pi \hat{\kappa})}{\pi \hat{\kappa} sin(\hat{\kappa}+t_2) - t_2 sin(\pi \hat{\kappa})} \qquad\qquad (2),$$

where ^ indicates an estimate of the distribution parameter. Further details on L-moments and their application to
distribution fitting are provided by Hosking and Wallis (1997) and Asquith et al., (2017). The GLO distribution
can be used to calculate a flood, $Q_T$, with a recurrence interval of $T$ years as

$$Q_T = \xi \left[ 1 + \frac{\beta}{\kappa} (1 - (T-1)^{-\kappa}) \right] = \xi z_T \qquad\qquad (3),$$

where $z_T$ is the 'growth curve' at $T$. The Weibull and Log-Pearson Type III distributions are also three parameter
distributions, described fully by Asquith et al. (2017) and Hosking and Wallis (1997) who define the relevant L-
moments and parameter calculations. The Gringorten (Cunnane, 1978) plotting position (Eq. (4)) was used,





$$x_i = (i - 0.44)/(n + 0.12) \qquad (4),$$

where $x_i$ is the $i$th quantile of the distribution, $i$ being the rank of the annual maximum flood in a given year, and $n$ the total number of years in the record. This method allows estimation of an event of up to $(1.79n + 0.2)$ years return period (Stedinger et al., 1993). Figure 2 shows typical data sets and curve fits.

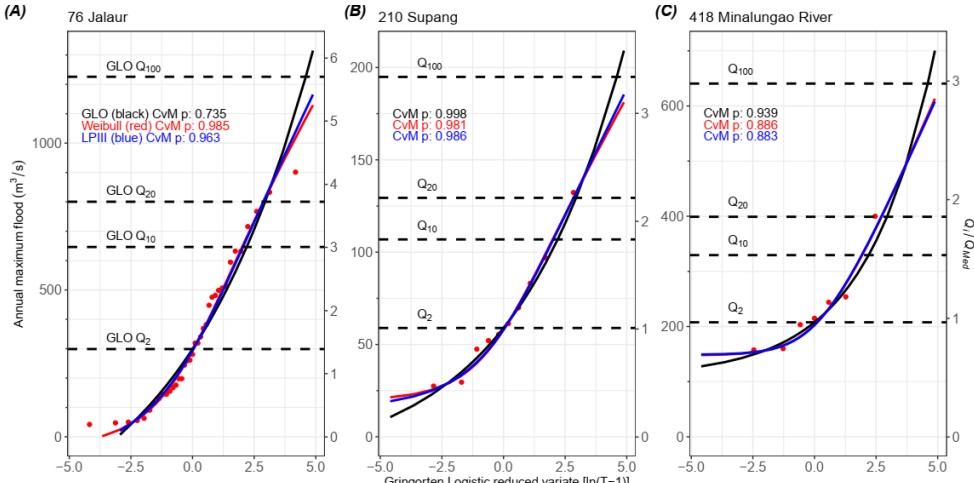

**Figure 2:** Selected annual maximum flood data and curve fits. Red points are data. Fitted curves are GLO (black),
Weibull (red) and Log Pearson III (blue). Cramer-von-Mises p-values shown. Left axes are flood magnitude
$(m^3.s^{-1})$ and right axes scale this by the median annual flood at each site. Values of 2,10,20 and 100 year recurrence
interval floods are indicated, calculated using the GLO method. (A) Site 76, Jalaur (Lat: 11.1195; Long: 122.5386;
Area 210km²; BRS_C data set; 37 years of data; best-fit curve: Weibull); (B) Site 210, Supang (Lat: 17.0073;
Long:120.9086; Area 56km²; Cagayan data set; 10 years; GLO); (C) Minalungao (or Sumacbao) River (Lat:
15.3430; Long: 121.0794; Area 309 km²; SWS data set; 7 years; GLO).

Analysis was undertaken in R (R Core Team, 2021), using the package lmomco (Asquith, 2020) to derive the L-
moment estimates, to fit the distributions and to calculate their significance. Of the 513 sites with records of at
least 7 years length, the minimum required for L-moment calculation, two had invalid L-moments and so are
excluded from further analysis. For these 513 sites, goodness-of-fit between the data and the three distributions
was assessed using Cramér-von Mises (CvM) test (Asquith, 2020). Such goodness-of-fit tests are unable to
definitively identify the best distribution to use, or if any of the distributions are adequate (Asquith, 2020),
particularly with relatively short records, as used here. Rather, the CvM p-values provide an indication of the
performance of the three distributions. The annual maximum series and the three curve fits were inspected for
each site and those with very poor fits were excluded. Mostly these excluded sites corresponded with low CvM
p-values, although this was not always the case. The distribution with the highest p-value from the CvM test was
used to provide $Q_x$ estimates for the site. This screening process led to the elimination of 205 sites from the data
set, leaving 466 that were further analysed. The distribution of the best-fit curves (Table 2) does not show
systematic differences between data source, catchment area or climate type (Table 2).





**Table 2** Best-fit curves, defined as those with highest Cramér-von Mises test p-value. 207 sites were excluded from the analysis, 2 due to L-moments not being valid, and the remainder due to having poor fit overall, based on the p-value and visual inspection.

| Best fit curve | All sites | Data Source BRS A/B/C | Cag | SWS | <100 | Catchment Area (km²) 100-199 | 200-399 | 400-799 | ≥800 | Climate Type I | II | III | IV |
|---|---|---|---|---|---|---|---|---|---|---|---|---|---|
| GLO | 184 | 99/52/6 | 13 | 14 | 58 | 39 | 31 | 21 | 35 | 48 | 21 | 66 | 49 |
| Weibull | 207 | 131/42/18 | 8 | 8 | 75 | 42 | 26 | 35 | 29 | 58 | 22 | 86 | 41 |
| Log Pearson III | 75 | 52/14/3 | 3 | 3 | 31 | 8 | 18 | 7 | 11 | 15 | 10 | 37 | 13 |
| TOTAL Used | 466 | 282/108/27 | 24 | 25 | 164 | 89 | 75 | 63 | 75 | 121 | 53 | 189 | 103 |
| Excluded – poor curve fit | 205 | 55/36/0 | 20 | 94 | 66 | 48 | 33 | 19 | 39 | 83 | 8 | 79 | 35 |
| L-moments not valid | 2 | 0 | 2 | 0 | 0 | 0 | 1 | 0 | 1 | 0 | 0 | 2 | 0 |
| TOTAL | 673 | 337/144/27 | 46 | 119 | 230 | 137 | 109 | 82 | 115 | 204 | 61 | 270 | 138 |

Values of $Q_2$, $Q_{10}$ and $Q_{100}$ were calculated from the fitted curves although the lengths of available records mean
that estimates of $Q_{100}$ are subject to significant uncertainty. Towards the high flow end of the data, the Weibull and Log-Pearson Type III curves are usually very similar, with the GLO curve typically being steeper and more curved (Fig. 2) and so providing higher flow estimates for high recurrence intervals ($Q_{20}$, $Q_{100}$) than the other two curves and often slightly lower estimates of $Q_2$ and $Q_{10}$. Ratios between flow estimates from different curves (Fig. S1) show this pattern: mean ratios between estimates from the GLO and Weibull distributions are
$Q_{2GLO}/Q_{2Wei} = 1.07$ (range 0.99 – 3.48), $Q_{10GLO}/Q_{10Wei} = 0.92$ (0.70-1.00) and $Q_{100GLO}/Q_{100Wei} = 1.09$ (0.42-1.15). Equivalent ratios for the GLO and Log-Pearson Type III curves are $Q_{2GLO}/Q_{2LPIII} = 1.10$ (1.00 – 4.27), $Q_{10GLO}/Q_{10LPIII} = 0.91$ (0.55 – 0.99) and $Q_{100GLO}/Q_{100LPIII} = 1.09$ (0.36-1.15). These ratios show some systematic differences between the distributions (Fig. 2, Fig. S1) and suggest that the choice of distribution influences flow estimates.

Estimating uncertainty in the $Q_x$ estimates is not straightforward (Kjeldsen, 2013; Kjeldsen and Jones, 2004) and reflects variability in the index flood, in the growth curve and in covariance between the index flood and the growth curve (Kjeldsen and Jones, 2004). For a single site, the factorial standard error for the GLO distribution, *fse*, is defined as (Kjeldsen, 2013):

$$fse = e^{\left(\frac{2\beta}{\sqrt{n}}\right)}$$ 
(5)

where $n$ is the number of annual maxima in the data set. However, derivation of Eq. (5) relies on approximations that limit the reliability of the equation when n ≤ 20 (Kjeldsen, 2013). On account of this, *fse* values were calculated only for records of at least 20 years length, all but one of which come from the BRS data sets (Table 1).





Growth curves were calculated for each of the 466 sites, using Eq.(3) and equivalents for the Weibull and Log-
Pearson Type III distributions, over the range of $-3.5 \leq \mathrm{Ln}(T\text{-}1) \leq 5.0$, i.e. return period $T$ in the range 1 to 149
years. Curves were standardised by dividing discharge by the median annual flood recorded at each site.

There are several ways to construct growth curves for: (i) each of the regions of the Philippines; (ii) each of the
four climate types (Fig. 1); and, (iii) for catchments of different areas, as identified in Table 2. Firstly, the curves
from each site within any of these groups can be combined, by calculating their mean, mean weighted by record
length, or median (Figs. S2-S4). Secondly, the data can be amalgamated for all sites within each group and GLO
curves fitted to the pooled data. The median and weighted mean methods lead to under-estimation of the longest
recurrence interval floods (Figs. S2-S4) whereas both the mean of GLO curves from each site and the GLO curves
fitted to the amalgamated data increase more rapidly at long recurrence intervals. Note that the variability between
sites within a region (or climate type or within catchments of similar area) provides an indication of the uncertainty
to be expected when using regionalised curves.

**4.2 Predicting high magnitude floods from catchment properties**

The values of $Q_T$ provided by the GLO analysis previously, were correlated with catchment properties. To apply
the FEH approach described earlier (Kjeldsen et al., 2008), catchment properties, precipitation and land use were
derived from a range of data sources. Table 3 summarises the variables used and provides a comparison with the
FEH method (Kjeldsen et al., 2008). Note that much of the data used are not contemporary and significant changes
in some variables, particularly land use but potentially also precipitation (Bagtasa, 2017), may have occurred since
the SWS data were collected in the early 20th Century.

**Table 3**. Variables used in the flood prediction analysis.

| FEH variable name | Units | FEH Definition | Philippines data equivalent | Variable names (this paper) |
|---|---|---|---|---|
| AREA | km$^2$ | Catchment area | Area from DEM of the catchment, calculated in ArcGIS | AREA |
| BFIHOST | - | Baseflow index from soil data | Excluded | - |
| DPLBAR | km | Drainage path length | Mean average drainage path length to catchment outlet for all segments of the stream network | DPLBAR |
| DPSBAR | m.km$^{-1}$ (FEH) m.m$^{-1}$ (this study) | Mean catchment slope | Mean average drainage path slope for all segments of the stream network | DPSBAR |
| EVAP | mm | Average annual potential evaporation | Excluded | - |
| FARL | - | Flood attenuation index (lakes etc) | Percentage/proportion of catchment area occupied by attenuation features (inland waters and fishing ponds) | ATT |
| FPEXT | - | Floodplain extent | Excluded | - |
| PRAT | none (FEH) mm (this study) | Ratio of $P_{100}/P_2$ for 1-day rainfall | Standard deviation of annual rainfall within the catchment from mean annual rainfall (1998-2015) APHRODITE dataset | RFSD |
| PROPWET | - | Proportion of time when soil moisture deficit <6mm | Excluded | - |





| | | | | |
|---|---|---|---|---|
| RMED | mm | Median annual maximum 1-day rainfall | Mean of maximum daily rainfall within the catchment from maximum daily rainfall (1998-2015) APHRODITE dataset | RMED |
| SAAR | mm | Annual mean rainfall 1961-90 | Mean of annual rainfall within the catchment from mean annual rainfall (1998-2015) APHRODITE dataset | SAAR |
| URBEXT2000 | - | Proportion of urban land cover in 2000 | Percentage of catchment area occupied by urban features (built-up) | URB |
| None | - | - | Percentage of catchment area occupied by agriculture (annual crop, fallow plus perennial crop) | AG |
| None | - | - | Percentage of catchment area occupied by closed and open forest | FOR |

National-scale catchment physical properties for the Philippines were previously calculated and are available as an open access geodatabase (Boothroyd et al., 2023). In brief, topographic analysis was undertaken using a digital elevation model (DEM) acquired in 2013 with a 5 m spatial resolution and 1 m root-mean-square error vertical accuracy (Grafil and Castro, 2014). The DEM was resampled to a 30 m spatial resolution in ArcGIS due to processing constraints. Here, AREA, DPLBAR and DPSBAR were extracted from the geodatabase. Rainfall data

were from the end-of-the-day adjusted version of the APHRODITE data set (V1901, Yatagai et al., 2012). Land use variables (ATT, URB, AG, FOR) were from the National Mapping and Resource Information Authority (NAMRIA) 2010 land cover data set (www.namria.gov.ph).

**5 Results**

**5.1 Flood magnitude estimation**

The L-moment ratio diagram (Figure 3; Figure S6) shows the relationship between L-skew and L-kurtosis differentiated by climate type and the optimal best-fit curve. Sites where each of the distribution types fit the data best do cluster close to the theoretical relationships for each of those distributions as expected. Neither climate type (Figure 3), data source, catchment area nor record length (Figure S6) show significant segregation on the L-moment diagram. Consequently, the 466 retained sites are considered as a single data set in subsequent analysis.




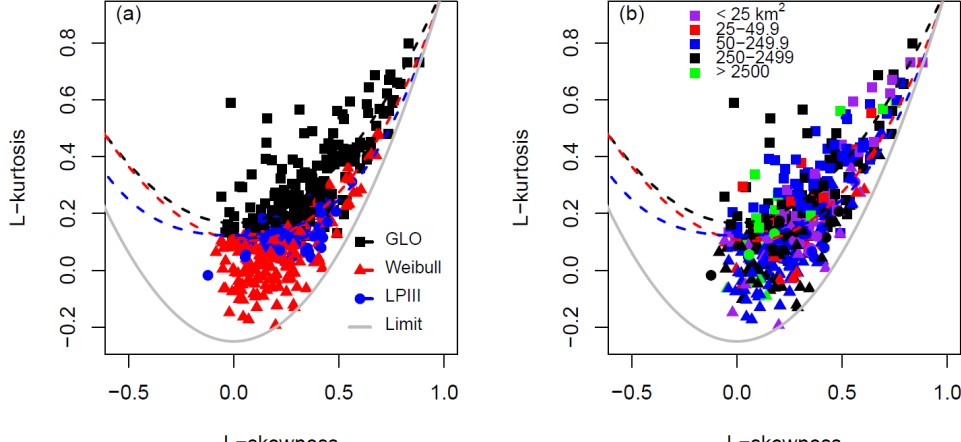

**Figure 3:** Relationships between L-skewness and L-kurtosis compared with theoretical curves (Hosking and Wallis, 1997). Data classified by: (a) best-fit curve; and, (b) catchment area. Fig. 3(a) shows segregation between sites with different best-fit curves, with higher positive L-kurtosis associated with the GLO curve, and low to negative L-kurtosis with the sites where the Weibull curve fits the data best. (b) shows overlap between the best-fit curve type and catchment areas with no clustering of different sized catchments. Colours indicate catchment areas, as shown at the top of the figure, and symbol shape indicates best-fit curve. Fig. S5 plots the data classified by climate type, length of record and data source: in all cases, there is no segregation according to the classifying variable.

The factorial standard error (*fse*) was computed using Eg. 5 for 71 sites with at least 20 annual maxima and for which the GLO distribution provided the best fit to the data. The values of *fse* range from 1.03 to 1.32, with mean = 1.18.

**5.2 Regional annual maximum daily flow growth curves**

Growth curves for all sites (Fig. 4a) show considerable variability within and between regions, reflecting the number, length and quality of available data records as well as catchment properties. Different climate zones (Fig. 4b) and catchment areas (Fig. 4c) indicate some grouping which may form the basis for hydrologic regionalisation. Climate types II and III plot higher than the others (Fig. 4b), although the median growth curves for all four climate types are very similar (Fig 4d). The pooled data provide steeper growth curves, reflecting the larger data series used and the increasing influence of large events in these larger samples. Consequently, the pooled data curves match high percentiles of the individual curves (shown by plotting close to, or sometimes outside of, the 75[th] percentile limits shown in Fig 4b,c). The steeper curves for pooled data are also seen when grouped according to catchment area (Fig 4e). Small (< 25 km$^2$) catchments plot separately from all larger areas, and there is little differentiation between any larger catchments. This contrasts with Meigh's (1995) results which suggested a steady decrease in $Q_x/Q_{mean}$ as catchment size increased.



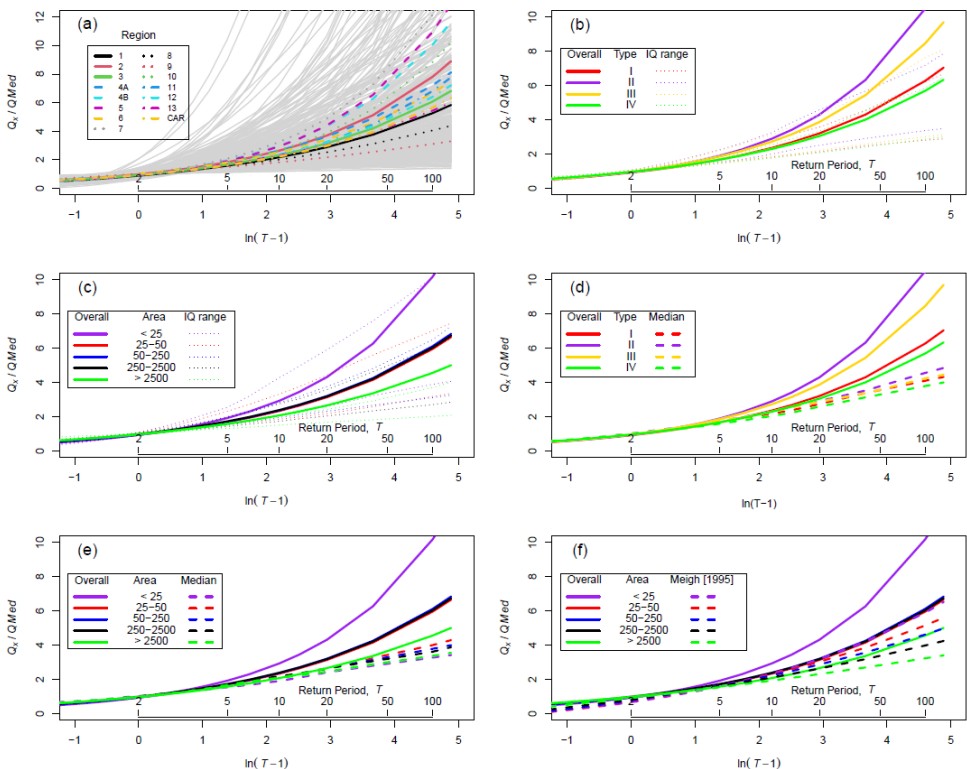

**Figure 4** Dimensionless growth curves. (a) individual curves for 466 sites, overlain by pooled curves for each region. (b) GLO curves for data pooled from all sites in each climate type; IQ range lines are the inter-quartile (25th and 75th percentiles) of the curves for individual sites within each climate zone. (c) GLO curves from bins of catchment area, with inter-quartile ranges from individual sites shown. (d) Comparison of GLO curves fitted to all data from within each climate zone and the median value from individual sites within that zone. (e) Comparison of GLO curves fitted to all data for sites within each catchment area bin and the median value from individual sites within that area bin. (f) Overall GLO curves for each catchment area bin, and adjusted equivalent curves from Meigh (1995). Adjustment was necessary as Meigh (1995) used the mean annual flood as the index flood, rather than the median. See text for details.

## 5.3 Flood estimation equations

### 5.3.1 Analysis of predictor variables

Each of the variables listed in Table 3, together with the $Qx$ estimates derived in the previous sections, were tested for normality and transformed as required (Table 4). Log10 transformation was used as the default, most variables being moderately positively skewed, with square-root transformation for two land-use and one rainfall variables that contained numerous zero values (areas of attenuation features and urban land, and standard deviation of rainfall). Cross-correlation plots and matrices, of the transformed variables where relevant, (Fig. S6) show expected autocorrelation between climate variables and no significant non-linear relationships elsewhere in the





predictor variables. Note (Table 4) that mean annual rainfall (SAAR) is poorly correlated with each of the *Qx*
measures.

**Table 4**. Summary statistics for variables used in the flood prediction analysis (466 sites). All values in original
units, prior to transformation (Trans). Land-use variables expressed as % were converted to proportion (0-1 scale)
for analysis. Correlation coefficient, R, significance: * p<0.01. Geometric mean (Geom mean) shown for variables
with no zero values. [+] one slope of 0.0 excluded when calculating geometric mean. $X_T$ = transformed value of
variable *X*.

| Variable (units) | Min / Max | | Mean | s.d. | Geom Mean / Median | Trans | R (log $Q_x$- $X_T$) | | | |
|---|---|---|---|---|---|---|---|---|---|---|
| | | | | | | | $Q_{MED}$ | $Q_2$ | $Q_{10}$ | $Q_{100}$ |
| AREA (km$^2$) | 1.13 27450 | / | 656 | 2040 | 172 / 163 | Log10 | 0.77* | 0.77* | 0.74* | 0.70* |
| DPLBAR (km) | 0.02 245.7 | / | 27.2 | 27.7 | 18.0 / 18.9 | Log10 | 0.74* | 0.74* | 0.71* | 0.67* |
| DPSBAR (m.m$^{-1}$) | 0.00 0.145 | / | 0.041 | 0.024 | 0.034[+]/ 0.044 | No | 0.03 | 0.03 | 0.07 | 0.10 |
| *ATT* (%) | 0 / 37.0 | | 1.11 | 2.4 | NA / 0.68 | √ | 0.34* | 0.34* | 0.30* | 0.28* |
| *RFSD* (mm) | 0; 444 | | 101 | 100 | NA / 78.0 | √ | 0.48* | 0.48* | 0.47* | 0.45* |
| RMED (mm) | 62.5 331 | / | 172 | 57.9 | 161 / 170 | No | 0.20* | 0.20* | 0.20* | 0.19* |
| SAAR (mm) | 1169 3877 | / | 2316 | 475 | 2269 / 2238 | Log10 | 0.06 | 0.06 | 0.05 | 0.03 |
| URB (%) | 0 / 51.3 | | 1.80 | 5.1 | NA / 0.48 | √ | -0.06 | -0.07 | -0.07 | -0.08 |
| AG (%) | 0 / 100 | | 36.9 | 27.6 | NA / 32.5 | No | -0.31* | -0.31* | -0.30* | -0.29* |
| FOR (%) | 0 / 86.4 | | 25.9 | 23.9 | NA / 19.2 | No | 0.28* | 0.28* | 0.29* | 0.29* |
| $Q_{MED}$ | 0.72 6029 | / | 380 | 722 | 132 / 136 | Log10 | - | 1.00* | 0.93* | 0.59* |
| $Q_2$ (m$^3$.s$^{-1}$) | 0.63 6211 | / | 374 | 717 | 131 / 141 | Log10 | - | - | 0.93* | 0.61* |
| $Q_{10}$ (m$^3$.s$^{-1}$) | 1.73 15230 | / | 831 | 1590 | 319 / 325 | Log10 | - | - | - | 0.82* |
| $Q_{100}$ (m$^3$.s$^{-1}$) | 3.75 91040 | / | 1801 | 5170 | 632 / 619 | Log10 | - | - | - | - |


### 5.3.2 Flood prediction from catchment area and rainfall

The correlations in Table 4 show that catchment area alone provides the most significant prediction of flood
magnitude. Drainage path length (*DPLBAR*) provides an equally good predictor as path length and catchment area
are correlated (Hack's law; Rigon et al., 1996). However, R$^2$ for catchment area and *DPLBAR* are in the range
0.45-0.6 so there is potential for additional variables improving flood magnitude prediction. Initially, the rainfall
variables were introduced to multiple regression relationships to account for the volume of water entering
catchments as catchment area * rainfall. Tables 3 and 4 show two relevant rainfall variables: *SAAR*, the mean
annual rainfall and *RMED*, the maximum daily rainfall which serves a measure of the magnitude of rainfall
extremes which may be expected to be correlated with flood peaks.

Equations using catchment area alone (Table 5) provide R$^2$ values between 0.49 ($Q_{100}$) and 0.6 ($Q_2$). These rise to
0.55-0.65 when area is multiplied by *RMED* (Table 5). $P_{99}$, the 99[th] percentile of daily rainfall, produces equations
which fit the data equally as well as *RMED*.





**Table 5**. Best-fit equations for the data set covering the whole of the Philippines (n=466). se = standard error of residuals.

| Event return period | Equations | $R^2$ | se |
|---|---|---|---|
| $Q_2$ | $Q_2 = 3.013A^{0.733}$ | 0.59 | 0.424 |
| | $Q_2 = 4.989 \times 10^{-2}(A.RMED)^{0.770}$ | 0.66 | 0.387 |
| $Q_{10}$ | $Q_{10} = 10.666A^{0.660}$ | 0.55 | 0.417 |
| | $Q_{10} = 2.576 \times 10^{-1}(A.RMED)^{0.696}$ | 0.62 | 0.383 |
| $Q_{100}$ | $Q_{100} = 25.645A^{0.622}$ | 0.49 | 0.442 |
| | $Q_{100} = 7.568 \times 10^{-1}(A.RMED)^{0.658}$ | 0.56 | 0.413 |


The residuals from the equations using *A.RMED* as the predictor were examined for effects of data source, climate type or region (Fig. 5). One-way ANOVA indicates significant differences between regions for $Q_2$, $Q_{10}$ and $Q_{100}$, with regions 7 (p=0.003; 0.0043; 0.026, respectively), 11 (p=0.012; 0.001; 0.005) and 12 (p<0.001 for all $Q_x$) being significantly different for all three return periods, region 3 (p=0.02; 0.02) for $Q_{10}$ and $Q_{100}$, and region 9

(p=0.02) for $Q_{100}$ only. Differences between climate types are only significant for $Q_{10}$ and $Q_{100}$, in both cases Type IV being significantly different from the others (p<0.01 in both cases). For data source, significant differences are noted for $Q_2$ and $Q_{10}$, in both cases due to BRS_B (p=0.006 for both) and the early 20[th] Century SWS (p<0.001; 0.014 for $Q_2$, $Q_{10}$, respectively) data sets. While these results suggest possible benefits from sub-dividing the data to produce predictive equations, inspection of Fig. 5, the boxplots and ANOVA results all show considerable

inter-group variance. Hence, the alternative approach of introducing additional variables to the analysis is considered as the next stage of the analysis, before regionalisation is considered in section 5.3.4.



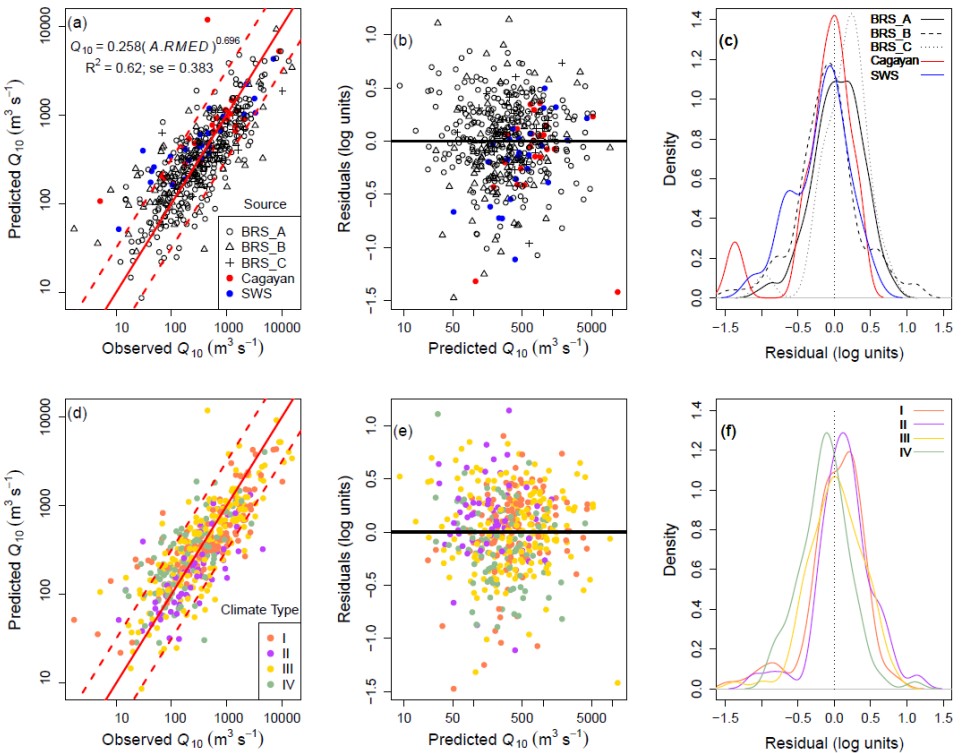

**Figure 5.** Observed values, prediction and residuals for $Q_{10}$ as a function of catchment area ($A$) multiplied by median daily maximum rainfall ($RMED$). (a)-(c) stratified by data source, (d)-(f) by climate type. (a),(d) are predicted vs. observed values, with 1:1 (solid), 1:2 and 2:1 (dashed) lines shown. Residuals (b) and (e) are normally distributed and show no systematic variation with predicted $Q_{10}$. Density plots of residuals (c), (f) confirm the absence of systematic variation with data source and climate type. Equivalent figures for $Q_2$ and $Q_{100}$ are in supplementary information (Figs. S7, S8).

### 5.3.3 Comprehensive stepwise regression prediction

Stepwise regression yielded equations (Eq. 6a-c) with between three and six significant (p<0.05) predictors, but overall R² values of 0.68, 0.63 and 0.57 for $Q_2$, $Q_{10}$ and $Q_{100}$, respectively. The modest improvements in R² associated with these additional variables suggest that there is limited value in using these complex equations for flood magnitude prediction.

$$Q_2 = 8.75 \times 10^{-3} A^{0.753} SAAR^{0.685} 10^{[0.002RMED - 2.423DPSBAR - 0.165AG - 0.676\sqrt{URB}]}$$

$[R^2 = 0.68; \text{rmse} = 0.377]$  (6a)

$$Q_{10} = 3.44(A)^{0.679} 10^{[0.003RMED - 0.75\sqrt{URB}]} \qquad [R^2 = 0.63; \text{rmse} = 0.378]  (6b)$$

$$Q_{100} = 8.49(A)^{0.667} 10^{[0.003RMED - 0.838\sqrt{URB} - 0.673\sqrt{ATT}]} \qquad [R^2 = 0.57; \text{rmse} = 0.407]  (6c)$$





This limitation is enhanced by consideration of the variables in the equations. Each equation contains land-use variables ($ATT$,$URB$,$AG$) that are determined from modern conditions. The relevance of these values to historical
data is uncertain given historic and contemporary land-use change across the Philippines. Their inclusion in equations for all three return periods does suggest that land-use may play a significant role in flood magnitude. In all three cases, $AREA$ enters the equation first, followed by $RMED$. $R^2$ values after each of these steps, for $Q_2$, $Q_{10}$ and $Q_{100}$ are: $AREA$ 0.59, 0.55, 0.49; and, $AREA$ and $RMED$ 0.66, 0.62, 0.55. Adding further variables (Eq. 6) improves $R^2$ by $\leq 0.02$, hence only catchment area ($AREA$) and median annual maximum daily rainfall ($RMED$)
are considered necessary for developing predictive equations. Whether these two predictors are added sequentially or are multiplied together (Table 5) does not affect overall model performance (note that the rmse values quoted in the equations are for the transformed variables). Subsequently, the product $AREA.RMED$ is used as a single measure of flood event rainfall volume across the catchments.

### 5.3.4 Regionalisation of predictive equations

The dimensionless growth curves (Fig. 4a), inspection and ANOVA analysis of regression residuals suggest that regionalisation may be able to improve predictive equations. Although the growth curves also show some segregation between climate types, this is not found to be a significant cause of variation in the residuals from predictive equations. Fitting equations to each region separately (Fig. 6a) yields improvement in $R^2$ and residual standard error for some regions, but this is inconsistent. The regional equations suggest that some grouping of
regions may be beneficial.

Three ways of dividing the 15 regions into groups were considered: (a) classification by visual inspection of the growth curves; (b) K-means cluster analysis of the intercepts ($a$) and gradients ($b$) for regression equations (Fig. 6a); and, (c) regionally contiguous groups used by Meigh (1995). Each grouping was tested for $Q_2$, $Q_{10}$ and $Q_{100}$ predictions. Results were consistent between these return periods, and results for $Q_{10}$ are given in Table 6 (see
Supplementary Information for $Q_2$ and $Q_{100}$ results).

**Table 6.** Equations for different groups of regions. Results for $Q_{10}$ are presented. Meigh (1995) did not include regions 13 or CAR, so the total number of sites in the three groups is 431.

| Group | Regions in group | Number of sites | Equation | $R^2$ | se |
|---|---|---|---|---|---|
| Growth curve | | | | | |
| A | 1,13,CAR | 65 | $Q_{10} = 0.234(A.RMED)^{0.730}$ | 0.78 | 0.245 |
| B | 2,3,4A,6,11,12 | 241 | $Q_{10} = 0.0945(A.RMED)^{0.779}$ | 0.64 | 0.390 |
| C | 4B,5,7,10 | 126 | $Q_{10} = 1.303(A.RMED)^{0.530}$ | 0.36 | 0.427 |
| D | 8,9 | 34 | $Q_{10} = 0.628(A.RMED)^{0.603}$ | 0.69 | 0.211 |
| K-means clustering of regional regression equations | | | | | |
| E | 1,6,7,8,11 | 142 | $Q_{10} = 0.095(A.RMED)^{0.796}$ | 0.75 | 0.298 |
| F | 2,3,4A,CAR | 167 | $Q_{10} = 0.071(A.RMED)^{0.813}$ | 0.69 | 0.389 |
| G | 4B,9,10,12,13 | 103 | $Q_{10} = 1.24(A.RMED)^{0.534}$ | 0.50 | 0.370 |
| H | 5 | 54 | $Q_{10} = 5.10(A.RMED)^{0.388}$ | 0.19 | 0.475 |
| Meigh (1995) contiguous regional groups | | | | | |
| I | 1,2 | 86 | $Q_{10} = 0.166(A.RMED)^{0.753}$ | 0.63 | 0.357 |
| J | 3,4A,4B,5,6,7,8 | 264 | $Q_{10} = 0.334(A.RMED)^{0.674}$ | 0.56 | 0.402 |
| K | 9,10,11,12 | 81 | $Q_{10} = 0.851(A.RMED)^{0.535}$ | 0.45 | 0.331 |



The R² and standard errors of residuals in Table 6 are compared with the combined results for all regions in Table

5 (R² = 0.62; se = 0.383). Weighting both the R² and residual error values by the number of sites in each
group/region suggested that for $Q_2$, $Q_{10}$ and $Q_{100}$ the highest R² values are those obtained using the overall
regressions on the full data set (Table 5). The residual standard errors are slightly lower when obtained from the
15 individual regional curves (0.36, 0.35, 0.37 for $Q_2$, $Q_{10}$ and $Q_{100}$, respectively) than from the overall regressions
(0.39, 0.38, 0.41). However, these differences are small and there is insufficient evidence to justify use of either

360    curves for individual regions or groups of regions.

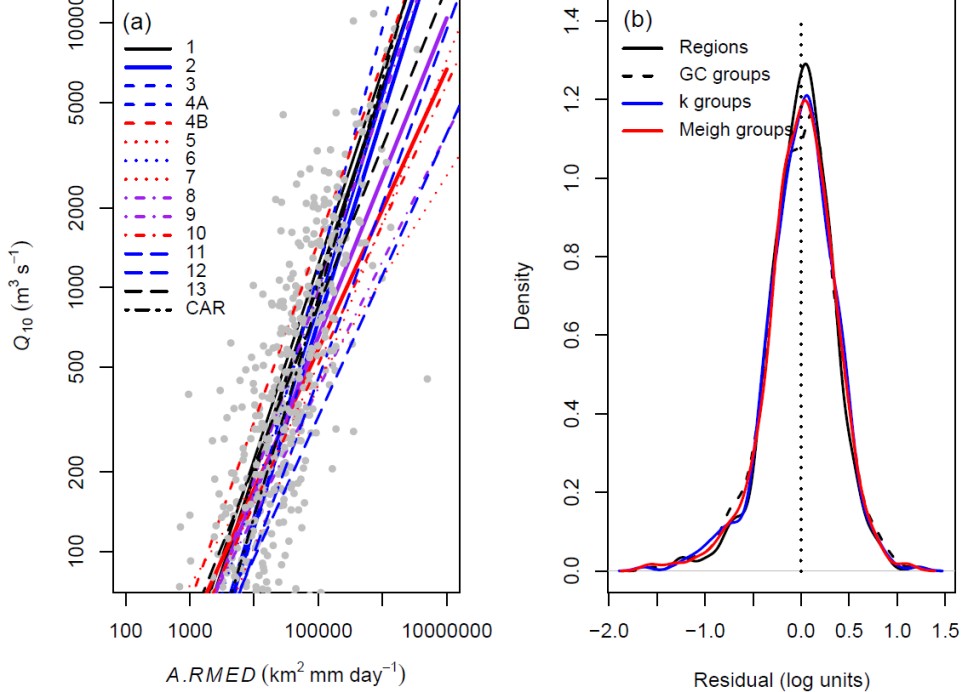

**Figure 6.** (a) Regression curves for each region in the form $Q_{10} = a\,(A.RMED)^b$. Curves are grouped according to
growth curve shapes (Table 6): group A (black), B (blue), C (red) and D (purple), and bold lines are regional
curves given by the equations in Table 6.  (b) Probability density functions for residuals from the individual

365    regional curves in (a), and the three groupings of regions in Table 6 (GC = Growth Curve; k = k-means). Note the
similarity in the distributions of residuals, although those for the individual regions are clustered slightly more
closely around the mean than those from the grouping methods.

**5.3.5 Spatial distribution of flood magnitudes and residuals**

The spatial distribution of calculated specific flood magnitudes ($Q_{xx}$ divided by catchment area $A$) (Figure 7a)

370    show a concentration of higher values through the central Philippines, with relatively lower values in NE Luzon
and across Mindinao in the south.  The underlying annual rainfall map shows a general decline from east to west,
and some of the highest rainfall areas are associated with high $Q_{xx}/A$ values, for example in the Bicol region.
Residuals from the overall equations (Table 5) do not show strong regional trends, although there are clusters of





positive and negative residuals in different regions. The residuals are not correlated with catchment area (R = -
0.04; p = 0.39) and only weakly with annual rainfall (R = 0.15; p < 0.001). However, there is a significant positive
correlation between residuals and specific flood magnitude (R = 0.62; p < 2 x $10^{-16}$), with only negative residuals
for $Q_{10}/A < 0.46$ and only positive residuals when $Q_{10}/A > 6.4$. These results are replicated for $Q_2$ and $Q_{100}$, with
significant correlations of 0.6 (p < 2 x $10^{-16}$) for both $Q_2/A$ and $Q_{100}/A$.

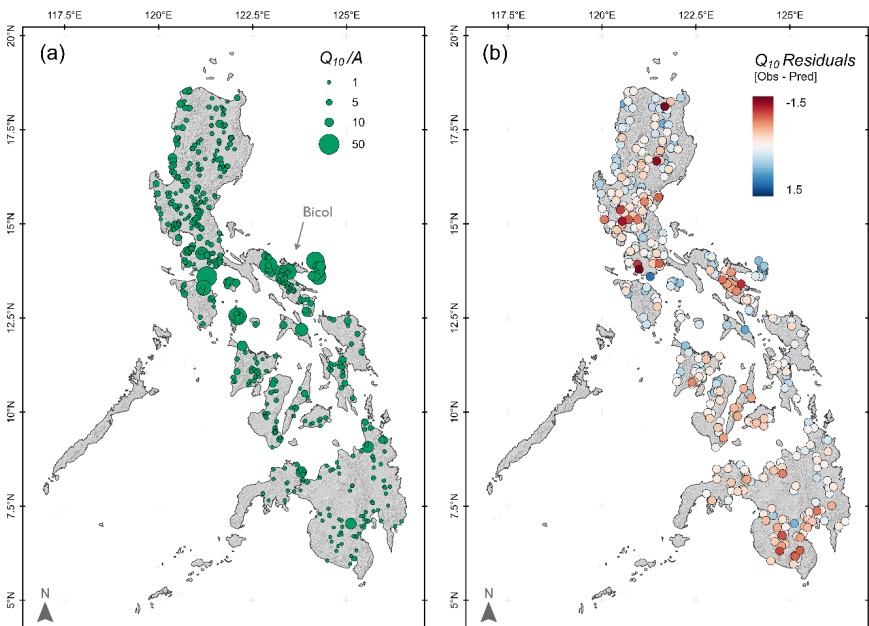

**Figure 7.** (a) Specific 10-year flood discharge ($Q_{10}/A$), showing generally higher values in the central Philippines
and southern Luzon, and lower values across Mindanao. (b) Residuals (in log10 units) from Philippines-wide
(Table 5) equations for $Q_{10}$. Note the absence of regional trends, although there are some sub-regional clusters of
both positive and negative residuals.

## 6 Discussion

### 6.1    Design equations for the Philippines

#### 6.1.1    Data availability and quality

Flow data were combined from four data sets that are partly independent, having been collected by different
agencies and using different methods, but which overlap significantly in collecting data at the same or nearby
locations. Catchment properties, such as area and gradients, were derived from a high resolution DEM that covers
the whole of the Philippines. Although some station locations are ambiguous in the data records, the locations of
all stations included in the analysis have been reliably identified using the descriptions in the original data sources.





Land use data rely on a single time, and no historical land use data are available. This introduces uncertainty to the analysis, especially for data collected a century or more prior to the land use data in areas that have undergone urban development and forest replacement by agriculture.

The proportions of variance in flood estimates that are statistically explained by the best-fit equations ($R^2$; Tables
5,6; Eq. 6) are within the range from studies in other tropical regions (Meigh et al., 1997), from 0.38 (Malawi) to 0.92 (Papua New Guinea). The relatively low $R^2$ values reflect a range of factors, including: data quality and length of flow records; changing climate and hydrological conditions during the time period covered by the study; and, controls over flood magnitude in these tropical catchments being influenced by hydrological parameters that are not considered in the analysis. Data quality has been assessed throughout, with sites excluded if their growth
curves are based on short records or do not fit expected shapes (Tables 1,2). Further, there is no evidence of bias in the data, shown both by the original variables and the behaviour of residuals from the final predictive curves. For example, the best-fit curves are not biased by data source, climate type or record length (Figures 3; S6,S8,S9). The residuals show neither systematic variation across these same categories (Figure 5) nor consistent spatial dependence (Figure 7).

Some spatial dependence is visible in Figure 7, although attempts to produce regionally consistent predictive curves (Table 6; Figure 6) do not improve the overall performance of the equations compared with national equations. The residuals in Figure 7 do not correlate clearly with either total rainfall (Figure 1B) or the relative importance of tropical cyclones in generating precipitation (Figure 1C). Further analysis of the role of regional climate in flood generation may be able to provide some improvements to predictions, although this is complicated
by ongoing climate change and potential changes in the importance of cyclonic precipitation (Bagtasa, 2017).

### 6.1.2   Recommended design equations

Neither the addition of further catchment variables (Eq. 6), nor regionalisation (Table 6) generated significant improvement in the predictive capabilities of the discharge equations. Hence, it is recommended that single national equations are utilised. This approach has the advantage of maximising the size of the data set used in
generating the equations; particularly for the largest catchments, the small sample size reduces confidence in the predictions in some regions. Regionally grouped equations (Table 6) can provide additional estimates of flood magnitude that may be helpful in some cases.

The recommended design equations for $Q_2$, $Q_{10}$ and $Q_{100}$ are those for the whole of the Philippines given in Table 5. Using only catchment area, $A$, will provide usable flood magnitude estimates, the uncertainty of which can be
estimated from the residual standard errors given in Table 5. Here we obtained *RMED* values from the APHRODITE database. *RMED* can be determined in other ways, and the sensitivity of flood predictions to changing RMED can be assessed directly. Along with catchment area, other catchment properties that provide information to contextualise the flood magnitude estimates can be obtained from an open access database (Boothroyd et al., 2023). Utilising design equations based on catchment area alone has the advantage of simplicity
of computation, but the relatively low $R^2$ values (Tables 5,6) obtained suggest that a simple multivariate regression approach offers only partial improvement to the predictive capability of the equations.





### 6.2 Comparison with other estimates

#### 6.2.1 Comparison with similar approaches

The previous large-scale study of Philippines flood magnitude (Meigh, 1995; Meigh et al., 1997) used a smaller data set than here, based mainly on BRS data from before 1980, and fitted only the General Extreme Value distribution to the annual maxima time series. The overlap in data means that Meigh's (1995) study cannot be considered to be independent of the present analysis and so does not provide a validation of our results. Some comparison between the two studies is valuable to illustrate the effects of using an expanded data set and the GLO

fitting approach (Figure 4f). Liongson (2016) used data from 29 stations and found that $Q_m = 5.90A^{0.763}$ ($R^2$=0.65), which is consistent with results in Table 5 as $Q_m$ lies between $Q_2$ and $Q_{10}$.

Meigh et al. (1997) present global data, although with an emphasis on tropical regions. Their best-fit equations contain few variables, often only catchment area with mean annual rainfall as the secondary predictor. Comparison of equations between sites revealed the expected overall pattern of higher specific discharges in more humid areas

with steeper growth curves in more arid locations that have more variable rainfall, as also seen in the data of Loebis (2002). The consistency of rainfall across the Philippines leads to a clear catchment area effect (Figure 4f) in growth curves for small (<25 km$^2$) and large (>2500 km$^2$) catchments, although using aggregated data shows no differentiation for catchments of intermediate sizes. Individual catchment growth curves show considerable variation within all of the catchment area bins, suggesting that caution is needed in using the aggregated curves

for predictive purposes at individual sites. Figure 4 provides a range of aggregated growth curves that can be applied according to catchment area and/or climate type. The differences between the median and mean curves on Figure 4 reflect skew in the growth curve distributions, which is likely to result from the use of relatively short records some of which will include long return period events so overestimating flood magnitudes. Median curves (climate type - Figure 4d; catchment area – Figure 4e) can be used in flood estimation, with the associated mean

values and inter-quartile ranges (Figure 4b,c) giving indications of the possible variability, and hence uncertainty, associated with these estimates.

#### 6.2.2 Comparison with rainfall-runoff modelling

The Philippines "Nationwide Disaster Risk and Exposure Assessment for Mitigation (DREAM) Program" produced reports for major Philippines river basins (https://dream.upd.edu.ph/products/publications/index.html)

that included flood magnitude estimation. In the DREAM study, 24-hour rainfall events with a range of return periods were calculated from data and these events were then used to model river flows in HEC-HMS 3.5 software. Comparisons are made using catchment area equations (Table 5) for $Q_{10}$ and $Q_{100}$ for sites with unambiguous locations from where DREAM results are reported and for which we are able to calculate catchment areas.

$Q_{10}$ and $Q_{100}$ comparisons (Figures 8a, S10) cluster around the 1:1 line of agreement. The HEC-HMS estimates

exceed the predictions using catchment area at 27 of 38 sites for $Q_{10}$, and at 24 sites for $Q_{100}$. Mean ratios between HEC-HMS and predicted values are 1.61 for $Q_{10}$ and 1.76 for $Q_{100}$. The HEC-HMS results are for instantaneous flows which will be greater than the predicted daily mean flows, with the magnitude of this difference depending on hydrograph shape and hence catchment size (Figure 8b). Given the uncertainties in the data and predictions noted above, and the limited calibration data available for the flood modelling in the DREAM project, the results

shown in Figure 8 provide confidence in both the HEC-HMS modelling undertaken for the DREAM project and
the catchment area based predictions developed herein.

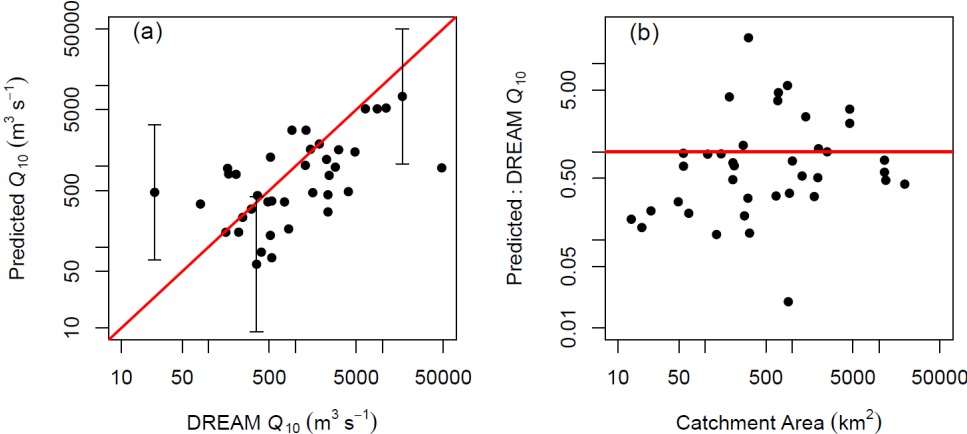

**Figure 8.** (a) Comparison between $Q_{10}$ estimates based on catchment area (Table 5) and HEC-HMS estimates
from the DREAM project. Red line is 1:1 equivalence. (b) Effect of catchment area on the ratio between $Q_{10}$
values from this paper and the DREAM HEC-HMS modelling. Red line shows equal $Q_{10}$ values from both
methods. DREAM estimates are instantaneous peak flows whereas the estimates herein are daily means. As
catchment area increases, equivalence between the two methods would show the $Q_{10}$ ratio increasing towards 1.0
as catchment area increases, with lower values in smaller catchments in which flood peaks are of much less than
one day duration. 95% prediction intervals are shown for selected points on (a) to indicate the magnitude of
statistical uncertainty in the predictions.  These are approximated as ±2s.e., where s.e. is the regression standard
error given in Table 5. Fig. S10 presents equivalent results for $Q_{100}$.

### 6.3    Combining data from multiple sources

Long hydrological time series are not commonly available worldwide, with particular challenges in developing
countries (Cabrera and Lee, 2020). More usually, short, discontinuous records are available and the challenge is
to make best use of these to produce regional or national design equations. Combining data from different sources
and that was collected over different time periods raises several issues, including: changing data gathering
methodologies; climate and land use changes; and, rating curve changes due to relocation of measuring sites
and/or river bed morphological change. Uncertainty in individual measurements was assessed here through careful
reading of available metadata and quality control. The metadata available for the early 20[th] Century SWS data
includes very detailed site descriptions, rating curves, assessment of site stability and statements on data reliability
from the authors (Irrigation Division, 1924). Such details are rarely available, at least in accessible public records,
for more recent data. The SWS reports provide useful insight into the challenges of hydrometric monitoring in the
Philippines, with several sites showing evidence of channel change and frequent shifts in rating curves. Although
beyond the scope of this paper, such changes in rating behaviour can be used to assess the impacts of land use and
climate changes on river sediment budgets (eg Slater et al., 2015).





The validity of combining data is difficult to assess directly. The residuals from predictive curves (Figure 5c), and similar disaggregation by data source for other parts of the analysis herein, show no significant difference between data sources. This absence of evidence of systematic bias between the data sources supports their aggregation. However, aggregation does need to be undertaken carefully with assessment of data quality and comparability at all stages of the analysis.

### 6.4     Enhancing the predictions

Tropical cyclones generate many of the significant floods in the northern Philippines, where they contribute over 50% of total rainfall (Figure 1; Bagtasa, 2017), but are very infrequent south of 10°N. Annual rainfall totals show less variability (Figure 1), although rainfall seasonality varies between climate types. Climate models predict increasing flood magnitudes across the Philippines north of 10°N for nearly all scenarios, with smaller or no increases predicted in southern regions (Tolentino et al., 2016).

The existing flow data base, coupled with geospatial information (Boothroyd et al., 2023), can be used for further analysis. Regional spatially-weighted grouping methods (Bocchiola et al., 2003; Griffiths et al., 2020; Muhammad and Lu, 2020) may reveal sub-regional controls over flood magnitude that will be able to improve predictions. Hydrological similarity between catchments does not necessarily imply regional proximity. In the Philippines, climatic gradients are observed both east-west due to topographic influences and north-south as a result of typhoon locations (Figure 1). Coupled with topographic diversity due to the range of island sizes and relief, a range of hydrological characteristics is expected across the country. Hence, statistical grouping (eg clustering, Figure 7) of catchments is necessary to identify hydrologically similar behaviour and provides a more cost-effective and achievable approach than resource-intensive rainfall-runoff modelling (Griffiths et al., 2020). Regional studies from the Philippines have shown the relative contributions that rainfall and topographic factors make to flood magnitude (Cabrera and Lee, 2020) and this approach may be extended nationally.

The methods in this study assume stationarity in the data time series, which has increasingly been questioned as the impacts of recent climate change and a range of anthropogenic factors on flood properties have been observed (Kalai et al., 2020; Kundzewicz et al., 2017). Consequently, approaches that explicitly consider non-stationary time series (eg. François et al., 2019; Kalai et al., 2020) are being developed and refined. Spatially variable responses to changing climate suggest the need for spatio-temporal modelling (Franco-Villoria et al., 2018) and regional calibration of predictive equations (e.g. Griffiths et al., 2020). Our combined data set will enable some of these analyses to be undertaken in the Philippines, so potentially improving the understanding and prediction of flood peaks.

### 7 Conclusions

Collation of historical data from multiple sources is a widely used technique in climatological and hydrological studies to extend modern records. Changes to data collection methods, to the environment in which the data are collected and to the ways in which data are recorded and reported all affect the reliability of such consolidated data sets. Here we access an extensive and well documented data set from the early 20th Century (SWS data; Irrigation Division 1923-24) that extends annual maximum flood records from the Philippines. The data set is extended from that analysed by Meigh (1995), although the results herein are largely consistent with that study.





Recent high-quality supplementary data on catchment properties, precipitation and land use have been added enabling assessment of a range of controls over flood magnitude.

Multivariate analysis shows that predictive equations for floods of recurrence intervals from 2 to 100 years based on catchment area alone have $R^2$ values no greater than 0.59, but that incorporating *RMED*, the median annual maximum 1-day rainfall, as a precipitation variable only increases $R^2$ to between 0.56 for $Q_{100}$ and 0.66 for $Q_2$. Very few other variables were significant when added to multiple regression equations. The relatively low $R^2$ values are typical of studies from tropical regions, suggesting that the Flood Estimation Handbook approach

developed for temperate climates requires some re-design for application to the tropics. The equations developed herein are suitable for use as design equations for the Philippines, but the uncertainties in predictions need to be assessed. Comparison with previous, independent, HEC-HMS modelling is encouraging but serves to illustrate the uncertainties in flood magnitude prediction that remain using either of these methods.

The Philippines exhibits regional climate variability, and there is some spatial structure in residuals from the

predictive equations. However, region-specific predictive equations do not perform significantly better than the national equations.

This study demonstrates the potential for combining data from multiple sources to generate flood magnitude predictions. Combining individually short records, after careful screening and exclusion of erroneous data, generates large data sets that can produce consistent results. Enhanced data gathering and extension of continuous

flood records are required to reduce uncertainties and improve flood forecasting, but the consistency across the Philippines suggests that extrapolation from a small number of carefully selected catchments could provide nationally reliable predictive equations with uncertainties that are considerably reduced from our results.

**8 Data availability**

The data that support the findings of this study are openly available in University of Glasgow's Enlighten Research

data repository at DOI: 10.5525/gla.researchdata.1666.

**9 Supplement link: the link to the supplement will be included by Copernicus, if applicable.**

**10 Author contribution**

Conceptualisation: TH, EP; Funding acquisition: RW, EP, TH, PT; Methodology: TH; Investigation: TH, PT, EG, JP, RB; Writing Original Draft: TH; Writing – Review and Editing: RW, PT, RB, CD, EP

**11 The authors declare that they have no conflict of interest.**

**12 Acknowledgements**

Funding was through grants from the Newton Fund of the UK Natural Environment Research Council (NERC; NE/S003312) and the Department of Science and Technology - Philippine Council for Industry, Energy and Emerging Technology Research and Development (DOST-PCIEERD), and from the Scottish Funding Council

Global Challenges Research Fund. P.L.M.T. is in receipt of a DOST-Science Education Institute (DOST-SEI) and British Council award.





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
