# Peer review of "Integrating historical archives and geospatial data to revise flood estimation equations for Philippine rivers"

_Hydrology and Earth System Sciences, 2024_

## Author Comment (AC1)

**Responses to comments: Anonymous Referee 2**

The paper by Hoey et al., entitled 'Flood estimation for ungauged catchments in the Philippines' aims at delivering design equations to estimate flood magnitudes and frequency in ungauged catchments across the Philippines.

The paper addresses an interesting topic, the design of equations to predict floods in data scarce region. The authors do enormous work to analyse the data. However, the methodology is not well defined/structured. Overall, the paper lacks clarity and needs to be restructured for better clarity.

**General comments:**

The title of the paper does not reflect the content of the paper appropriately. The authors propose and evaluate the index flood approach and multi-variate regression to estimate flood. The approach is evaluated only at catchments with data used to fit the regression equations. In a context of flood estimation in ungauged catchments, the paper is lacking the following steps: 1) a clear definition of the methodology used for regionalization to ungauged catchment, 2) a cross-validation to evaluate the performance of the designed equations in ungauged catchments. It is necessary to evaluate how well the proposed approach would perform in ungauged catchments. Therefore, I suggest the authors consider using some cross-validation approach (i.e., leave-one-out or k-cross validation) where a set of catchments are used to fit the design equations; then the remaining catchment(s) are used as pseudo-ungauged catchment(s) to evaluate the accuracy of the designed equations in ungauged catchments. In addition, the methodology used is not clearly described (see specific comments below).

Response: These are all very good suggestions, based on the title of the paper. As noted in our response to Referee 1, this title is not accurate and we propose to amend the title to fit the contents of the paper. The reviewer's suggestions are exactly what we would do if the paper was to retain its current title, and there is merit in presenting such an approach and flow chart to guide readers in the use of the equations. However, that would lengthen the current paper excessively and we consider that this is a separate piece of work.

The paper could benefit from a flow diagram that describes the steps followed to estimate floods at ungauged catchments.

Response: This could be added as part of a short (approx 400 words) new section 6.5 'Applying the equations to ungauged catchments', although this would extend the paper. We would welcome the editor's view on this addition – it would be easy to produce, but would extend an already lengthy manuscript.

**Specific comments:**

- The authors fit and compare the accuracy of the three distributions showing that the choice of the distribution influences flow estimates and no distribution performs well at all sites. Then, the GLO distribution is selected to predict high flow magnitude in section 4.2 with no justification of this choice.

  Response: There is an error in line 215, in that the $Q_T$ values used were from the **best-fit curve** at each site, and not only from the GLO results. This will be corrected.

  In addition, Equation 5, the factorial standard error for the GLO distribution, is only applicable when number of records is at least 20 years. The study region has only 71

sites (line 245), with more than 20 years data, among the 466 sites retained for the analysis (Line 234) . Why choose the GLO distribution if it cannot be applicable to all sites? Why choose only GLO instead of using the best fitted distribution at each site as shown in table 2?

Response: See previous comment – clarifying the curves used will address all of these points.

- The last paragraph in section 4.1, the authors lists three ways the construct growth curves by combining the curves fitted from each site by region, climate type or catchment areas. Therefore, I am assuming that the resulting curves are regionalized curves where catchments from within each group(region or climate type, catchment area bin) would be represented by a single curve. Correct In other words, catchments of each group would have a single $Q_T$ value estimate from the regionalized growth curve. How the author use these results to perform a correlation analysis with catchment properties in Section 4.2 (Line 212:   values of $Q_T$ provided by the GLO analysis previously, were correlated with catchment properties")?

Response: The analysis reported in 4.2 is for all catchments individually, and does not used the regionalised growth curves.  Section 4.2 does not follow from the final paragraph in section 4.1 as the reviewer has assumed.  We will revise the wording in lines 205-213 to make it clear why we are producing combined growth curves and how these can be used, and can similarly edit line 215 to clarify that section 4.2 is concerned with individual catchments.

- The authors use precipitation dataset covering only 17 years (from 1998-2015) which is not sufficient to derive climatology of the catchments. In addition, this dataset are not available for a period that overlaps with flows records period. As suggestion, the author could use precipitation datasets from reanalysis products (e.g. ERA5) which are available from 1940 to present around the globe. It would cover the same period as flow records for most catchments and provide sufficient long timeseries to derive climatology for precipitation variables used in Table 3.

Response: This is an interesting suggestion. We are aware that none of the available precipitation datasets is perfect for our analysis, as we are combining river flow data from different time periods. We used APHRODITE V2 which, as noted, covers the period 1988-2015. Compared with the other precipitation reanalysis products that are available, APHRODITE has the advantage for our study of being focused on Asia and so is likely to be more reliable in that region. The V2 product is regarded as an improvement over APHRODITE V1 in its representation of extreme events which is another benefit for using it in our study. For both of these reasons, we are satisfied that this data set is appropriate, but note that a comparison between different precipitation estimates would be an interesting area for future work.

- Why the fitted curves from individual sites are not used to fit a relationship between flood magnitude and catchment area?

Response: Table 5 presents the results of these fits and compares them with equations that also include a precipitation term.

- It is unclear how the individual short historical records are combined to generate large dataset that produce consistent results.

Response: This is covered in section 6.3, and we can make it more explicit. In summary, the absence of systematic differences between results from the different data sets (eg Figure 5a-c) provides confidence that we can treat the data from all data sources as a single data set for the analyses presented in the paper.

- Results presented in Section 5.3.3 (Eq6a-6c) would be better presented in a table with equation and associated $R^2$ and RMSE in the same way as in Table 5.

    Response: That appears to be a good suggestion for consistency within the paper.

Overall, the way the methodology and the results are presented is incoherent and lacks clarity. Other than the specific comments above, it is difficult to know what the reviewer is referring to here. Both reviewers have made specific comments that will improve the clarity of the results. The mis-leading title of the paper may have caused some confusion, and clarifying this will hopefully also aid the clarity of the paper.

---

## Author Comment (AC2)

**Responses to comments: Anonymous Referee 1**

This study evaluates 11 physical variables for index flood estimation across catchments in the Philippines, aiming to enhance flood estimation for ungauged catchments. The authors show significant effort in data collection and selection, and they present extensive analyses in this manuscript.

Response: Comments noted and appreciated.

Notice that the authors claim their study is applicable to ungauged catchments. However, if my understanding is correct, the analyses presented do not show anything regarding this applicability. While they propose using more local information to improve flood estimation—a common approach in many studies—they suggest this could benefit ungauged sites. Although this suggestion might be correct, it is overstated in the title since there are no relevant analyses or validation to support this claim.

Response: We note and largely agree with these comments – the aim of the paper is to provide a methodology that **can** be applied to ungauged catchments rather than demonstrating its effectiveness in such situations. The title does overstate this applicability, and we propose to revise this to 'Integrating historical archives and geospatial data to revise flood estimation equations for Philippine rivers' or something similar. In addition, we will ensure that the potential applicability to ungauged catchments is stressed rather than illustrating this – we could add a further section to the paper to demonstrate how the method can work in ungauged catchments, but that would extend an already lengthy paper and would require considerable statistical work to provide robust evidence.

In addition, the manuscript has several critical issues regarding the quality: 1) Unclear Critical Information - There is confusion regarding the number and details of study sites, as well as incomplete or unclear descriptions of methodologies.

Response: See replies below to specific points raised

2) Lack of Novelty and Significant Findings - The framework lacks innovation and the findings are not particularly groundbreaking (as noted by the authors in line 527).

Response: This comment is to some extent true, but we contend that the novelty in the paper comes from: (i) combining data from multiple sources to extend the database in what is a relatively data-poor country – this is the first analysis adding data from the early 20th century (SWS in Table 1) that covers sites for which no later data are available; (ii) utilising modern databases, for topography, rainfall and land-cover, to significantly extend previous analyses of flood magnitude in the Philippines; and, (iii) assessing the results against those form a largely independent hydrological modelling exercise.  The consistency of our results with previous analyses reveals that the noise observed by previous authors (Meigh, 1995; Meigh et al., 1997) is not due to excluding precipitation and land-use variables from the analysis, but rather is a consequence of catchment and/or climate characteristics that remain unknown and worthy of further study. We contend that this finding is itself novel and represents progress in our understanding of flood generation in the Philippines.

3) Quality and Clarity - The structure of the manuscript, along with its figures and tables (including captions), lacks quality and clarity. There are numerous mistakes throughout the document. I found it challenging to understand the authors' main points, both from the text and the figures. While the authors' efforts in conducting numerous analyses are commendable, they are strongly encouraged to improve the manuscript by enhancing its accuracy, clarity, and focus.

Response: Comments below on specific issues raised, and the general points regarding clarity will be considered when revising the manuscript.

Some specific comments (but not all) for improvement are listed below for reference:

How many catchments are analyzed exactly? Is it 513 or 466? The abstract states 513, but other parts of the manuscript (e.g., Figure 1, Table 2, line 172) suggest it is 466. Lines 164-172 are particularly confusing: how does 513 minus 205 result in 466 sites?

Response: Noted – the steps by which the full data set was censored are explained in the text but this can be made more explicit and we note inconsistency between Tables 1 and 2 that can be corrected.  The '205' on line 167 is an error. The table below shows the process that was followed, with each row representing a step in the analysis that resulted in the exclusioin of some sites:

|  | Number excluded | Number remaining |
|---|---|---|
| Total number of sites | - | 842 |
| Exclude sites with poor rating curves or indeterminate location | 169 | 673 |
| Exclude sites with short records | 160 | 513 |
| Exclude sites with invalid L-moments | 2 | 511 |
| Exclude sites with poor curve fit | 45 | 466 |

The catchment area sizes are analyzed in this study, but this information is missing in the data section.

Response: The method for calculating catchment area is noted in the supplementary data file and will be added to the manuscript.

Why do the catchment area groupings differ between Table 2 and Figure 3? The former uses four groups (100-200, 200-400, 400-800, <800), whereas the latter uses five groups (<25, 25-50, 50-250, 250-2500, >2500), with so different ranges.

Response: We agree that this inconsistency is not helpful – Table 2 will be edited to match the groups used in the figures. Table 2 does have 5 groups, as there is a <100km$^2$ group also.

Is the area grouping range rational? Comparing catchments across such varied groupings (<25, 25-50, 50-250, 250-2500, >2500) seems to be strange for representing hydrological responses.

Response: Given the shape of the frequency distribution of catchment areas, a non-linear grouping is appropriate. We tested a strictly logarithmic division of catchment areas and found no appreciable differences to the results when using the proposed grouping.  The total number of catchments within each group is as follows (total = 466):

| Catchment area km$^2$ | <25 | 25-50 | 50-250 | 250-2500 | >2500 |
|---|---|---|---|---|---|
| Number of sites | 57 | 34 | 190 | 165 | 20 |

The presentation of results in catchment area groups allows readers to assess visually if there are catchment area related effects in the data. It is intended only as a visual device and so the choice of group boundaries is not in itself significant. A much more comprehensive assessment of the importance of catchment area is given by Figures 5 and 6, and Tables 5 and 6.

The title of Section 5.1 is not coherent with its content.

Noted and this will be changed.

Moreover, it is difficult to discern the patterns the authors aim to show (lines 232-234) because Figure 3 is unclear. Improving the color and marker settings or changing the plot type (e.g., stacked bar plots) might be helpful.

Response: We will experiment with smaller markers for clarity. The plot style is the standard format for these results.

Line 233: Figure 3 refers to area, not climate.

Response: Noted and will be corrected.

Line 233: All the others should point to Figure S5, not Figure S6.

Response: Noted and will be corrected.

There are references to regions 1-13 in many analyses, but no introduction or proper definition of these regions is provided. The first mention appears on line 78, without any definition.

Response: We will add a definition of the regions of the Philippines and an appropriate reference to section 3.

Line 213 claims that the authors apply the FEH approach described earlier, but there is no earlier description. The term is introduced on line 50, but without sufficient detail.

Response: This is a fair comment – we have referenced several FEH studies but do not provide a brief summary of the FEH approach. The best place to do this is in section 4.2, where our methods are related to the FEH approach.

Improper structure: Lines 220-227 fit better in the data section rather than the methods section.

Response: We disagree with this suggestion as the description of variables in 220-7 needs to follow Table 3. The title of section 4.2 is "Predicting high magnitude floods from catchment properties" which is entirely appropriate for defining data sources for these variables.

Line 245: Why are only 71 sites analyzed here instead of the full 466 sites?

Response: As it says in line 248 "…for 71 sites with at least 20 annual maxima and for which the GLO distribution provided the best fit to the data." We could perhaps try to clarify this statement with some minor re-wording for emphasis "…for the 71 sites that had at least 20…"

The figure captions should be more descriptive of the settings, but discussions of the results/patterns should not be included here.

Response: It is not clear what 'more descriptive of the settings' means. The figure captions include very little discussion of the results presented within them, and where this is presented we think that it is valuable to guide the reader's interpretation of the figure. Readers may look at figures in isolation and the captions have been prepared to allow figures to be understood without having to read all of the related text.

Tables could be improved by adding delineation lines, clarifying headers (e.g., the last item in the first column of Table 2), and removing unnecessary information.

Response: The journal format has been followed for the Tables, although the reviewer's comment about delineation lines does make sense. We have tried to keep tables concise and are unsure what 'unnecessary information' is being referred to here.

In summary, I acknowledge that such a study is needed for the selected country, as the authors claim there are no other similar studies to such an extent. While the analyses are comprehensive, the manuscript lacks sufficient clarity in its structure and critical information, which hampers its transferability and overall readability.

Response: Some of the specific comments made by both reviewers will definitely assist in this regard.

---

## Author Response (AR1)

20 December 2024

**Dr Theresa Blume** and **Professor Alberto Guadagnini** Chief Executive Editors Hydrology and Earth Systems Sciences**

Ref: hess-2024-188

Dear Editors,

Following receipt of review comments and your invitation to revise our manuscript, I am pleased on behalf of our research team to submit a revised version of our paper. The title of the paper has been changed, following reviewers' comments, to *Integrating historical archives and geospatial data to revise flood estimation equations for Philippine rivers*.

The marked-up version of the paper shows where changes have been made in response to the comments received. The following table shows our responses to the specific comments made by the two reviewers.

| Number        | Comment                                                                                                                                                                                                                                                                                                                                                                                                                                                                                                             | Response and revisions                                                                                                                                                                                                                                                                                                                                                                                                                                                                                                                                                                                                                                                                                                                                                                                                                                                      |
|---------------|---------------------------------------------------------------------------------------------------------------------------------------------------------------------------------------------------------------------------------------------------------------------------------------------------------------------------------------------------------------------------------------------------------------------------------------------------------------------------------------------------------------------|-----------------------------------------------------------------------------------------------------------------------------------------------------------------------------------------------------------------------------------------------------------------------------------------------------------------------------------------------------------------------------------------------------------------------------------------------------------------------------------------------------------------------------------------------------------------------------------------------------------------------------------------------------------------------------------------------------------------------------------------------------------------------------------------------------------------------------------------------------------------------------|
| Reviewer
1 |                                                                                                                                                                                                                                                                                                                                                                                                                                                                                                                     |                                                                                                                                                                                                                                                                                                                                                                                                                                                                                                                                                                                                                                                                                                                                                                                                                                                                             |
| R1.1          | This study evaluates 11 physical variables for index flood estimation across catchments in the Philippines, aiming to enhance flood estimation for ungauged catchments. The authors show significant effort in data collection and selection, and they present extensive analyses in this manuscript.                                                                                                                                                                                                               | Comments noted and appreciated.                                                                                                                                                                                                                                                                                                                                                                                                                                                                                                                                                                                                                                                                                                                                                                                                                                             |
| R1.2          | Notice that the authors claim their study is applicable to ungauged catchments. However, if my understanding is correct, the analyses presented do not show anything regarding this applicability. While they propose using more local information to improve flood estimation—a common approach in many studies—they suggest this could benefit ungauged sites. Although this suggestion might be correct, it is overstated in the title since there are no relevant analyses or validation to support this claim. | We note and largely agree with these comments – the aim of the paper is to undertake data analysis to generate equations that can be applied to ungauged catchments rather than demonstrating its effectiveness in such situations. The title, and abstract, did overstate this applicability and we have changed the title to 'Integrating historical archives and geospatial data to revise flood estimation equations for Philippine rivers' as well as editing the abstract. In addition, we have stressed the potential applicability of our results to ungauged catchments. We considered adding a further section to the paper to demonstrate how the method can work in ungauged catchments, but decided not to do this as it would extend an already lengthy paper and would have required considerable statistical work to provide robust evidence. |

| R1.3 | In addition, the manuscript has several critical issues regarding the quality: 1) Unclear Critical Information - There is confusion regarding the number and details of study sites, as well as incomplete or unclear descriptions of methodologies.                                                                                                                                                                                                                                   | See replies below to specific points raised. We have made some clarifications and have corrected some erroneous values in the text.                                                                                                                                                                                                                                                                                                                                                                                                                                                                                                                                                                                                                                                                                                                                                                                                                                                                                                                                                                                                                                                                                                          |
|------|----------------------------------------------------------------------------------------------------------------------------------------------------------------------------------------------------------------------------------------------------------------------------------------------------------------------------------------------------------------------------------------------------------------------------------------------------------------------------------------|----------------------------------------------------------------------------------------------------------------------------------------------------------------------------------------------------------------------------------------------------------------------------------------------------------------------------------------------------------------------------------------------------------------------------------------------------------------------------------------------------------------------------------------------------------------------------------------------------------------------------------------------------------------------------------------------------------------------------------------------------------------------------------------------------------------------------------------------------------------------------------------------------------------------------------------------------------------------------------------------------------------------------------------------------------------------------------------------------------------------------------------------------------------------------------------------------------------------------------------------|
| R1.4 | Lack of Novelty and Significant Findings - The framework lacks innovation and the findings are not particularly groundbreaking (as noted by the authors in line 527).                                                                                                                                                                                                                                                                                                                  | This comment is to some extent true, but we contend that the novelty in the paper comes from:  (i) combining data from multiple sources to extend the database in what is a relatively datapoor country and region – this is the first analysis adding data from the early 20th century (SWS in Table 1) that covers sites for which no later data are available; (ii) utilising modern databases, for topography, rainfall and land-cover, to significantly extend previous analyses of flood magnitude in the Philippines; and, (iii) assessing the results against those form a largely independent hydrological modelling exercise. The consistency of our results with previous analyses reveals that the noise observed by previous authors (Meigh, 1995; Meigh et al., 1997) is not due to excluding precipitation and land-use variables from the analysis, but rather is a consequence of catchment and/or climate characteristics that remain unknown and worthy of further study. We contend that this finding is itself novel and represents progress in our understanding of flood generation in the Philippines. Some editing of the abstract, introduction and discussion has been undertaken to try to clarify these points. |
| R1.5 | 3) Quality and Clarity - The structure of the manuscript, along with its figures and tables (including captions), lacks quality and clarity. There are numerous mistakes throughout the document. I found it challenging to understand the authors' main points, both from the text and the figures. While the authors' efforts in conducting numerous analyses are commendable, they are strongly encouraged to improve the manuscript by enhancing its accuracy, clarity, and focus. | Comments below on specific issues raised, and the general points regarding clarity have been considered in making revisions to the manuscript.                                                                                                                                                                                                                                                                                                                                                                                                                                                                                                                                                                                                                                                                                                                                                                                                                                                                                                                                                                                                                                                                                               |
| R1.6 | Some specific comments (but not all) for improvement are listed below for reference: How many catchments are analyzed exactly? Is it 513 or 466? The abstract states 513, but                                                                                                                                                                                                                                                                                                          | Noted – the steps by which the full data set was censored are explained in the text but has been be made more explicit and we note inconsistency between Tables 1 and 2 that has now been corrected. The '205' on line 167 in the original manuscript was an error. The table below shows                                                                                                                                                                                                                                                                                                                                                                                                                                                                                                                                                                                                                                                                                                                                                                                                                                                                                                                                                    |

|      | other parts of the manuscript (e.g., Figure 1, Table 2, line 172) suggest it is 466. Lines 164-172 are particularly confusing: how does 513 minus 205 result in 466 sites?                                                                   |                                                                                                                                                                                                                                                                                                                                          | the process that was followed, with each row representing a step in the analysis that resulted the exclusion of some sites: |                             |                           |                                     |                                 |                                |
|------|----------------------------------------------------------------------------------------------------------------------------------------------------------------------------------------------------------------------------------------------|------------------------------------------------------------------------------------------------------------------------------------------------------------------------------------------------------------------------------------------------------------------------------------------------------------------------------------------|-----------------------------------------------------------------------------------------------------------------------------|-----------------------------|---------------------------|-------------------------------------|---------------------------------|--------------------------------|
|      |                                                                                                                                                                                                                                              |                                                                                                                                                                                                                                                                                                                                          |                                                                                                                             |                             |                           | lumbe                               |                                 | umber
maining               |
|      |                                                                                                                                                                                                                                              |                                                                                                                                                                                                                                                                                                                                          | Total
of sit                                                                                                             | numbe
es                 | er -                      |                                     | 84                              | 2                              |
|      |                                                                                                                                                                                                                                              |                                                                                                                                                                                                                                                                                                                                          | with prating or indet                                                                                                       | g curve                     | es                        | 69                                  | 67                              | 3                              |
|      |                                                                                                                                                                                                                                              |                                                                                                                                                                                                                                                                                                                                          |                                                                                                                             | ude site
short
ds     | es 1                      | 60                                  | 51                              | 3                              |
|      |                                                                                                                                                                                                                                              |                                                                                                                                                                                                                                                                                                                                          | with i                                                                                                                      |                             | L-                        |                                     | 51                              |                                |
|      |                                                                                                                                                                                                                                              |                                                                                                                                                                                                                                                                                                                                          | with curve                                                                                                                  |                             | es 4                      | 5                                   | 46                              | 6                              |
| R1.7 | The catchment area sizes are analyzed in this study, but this information is missing in the data section.                                                                                                                                    | The method for calculating catchment area is noted in the supplementary data file and has been added to the manuscript.                                                                                                                                                                                                                  |                                                                                                                             |                             |                           |                                     |                                 |                                |
| R1.8 | Why do the catchment area groupings differ between Table 2 and Figure 3? The former uses four groups (100-200, 200-400, 400-800, <800), whereas the latter uses five groups (<25, 25-50, 50-250, 250-2500, >2500), with so different ranges. | We agree that this inconsistency is not helpful — Table 2 has been edited to match the groups used in the figures. Note that Table 2 does have 5 groups, as there is a <100km² group also.                                                                                                                                               |                                                                                                                             |                             |                           |                                     |                                 |                                |
| R1.9 | Is the area grouping range rational? Comparing catchments across such varied groupings (<25, 25-50, 50-250, 250-2500, >2500) seems to be strange for representing hydrological responses.                                                    | Given the shape of the frequency distribution of catchment areas, a non-linear grouping is appropriate. We tested a strictly logarithmic division of catchment areas and found no appreciable differences to the results when using the proposed grouping. The total number of catchments within each group is as follows (total = 466): |                                                                                                                             |                             |                           | is
mic
o
en using
er of |                                 |                                |
|      |                                                                                                                                                                                                                                              | Catchi
area k                                                                                                                                                                                                                                                                                                                         |                                                                                                                             | <25                         | 25-
50                 | 50-
250                          | 250-
2500                    | >2500                          |
|      |                                                                                                                                                                                                                                              | Number                                                                                                                                                                                                                                                                                                                                   | er of                                                                                                                       | 57                          | 34                        | 190                                 | 165                             | 20                             |
|      |                                                                                                                                                                                                                                              | The pres
groups a
are catc
is intend
choice o                                                                                                                                                                                                                                                                                | allows
hment
ed on                                                                                                    | reader
area r
ly as a | s to a
elated
visua | ssess
I effect
I devic        | visually
s in the
e and s | if there
data. It
to the |

|       |                                                                      | significant. A much more comprehensive                                                           |
|-------|----------------------------------------------------------------------|--------------------------------------------------------------------------------------------------|
|       |                                                                      | assessment of the importance of catchment area                                                   |
|       |                                                                      | is given by Figures 5 and 6, and Tables 5 and 6.                                                 |
| R1.10 | The title of Section 5.1 is not                                      | Noted and has been changed.                                                                      |
| R1.11 | coherent with its content.                                           | Ma have used smaller magnitude. The plat style is                                                |
| KI.II | Moreover, it is difficult to discern the patterns the authors aim to | We have used smaller markers. The plot style is the standard format for these results and so has |
|       | show (lines 232-234) because                                         | been retained.                                                                                   |
|       | Figure 3 is unclear. Improving the                                   | Sour rotalised.                                                                                  |
|       | color and marker settings or                                         |                                                                                                  |
|       | changing the plot type (e.g.,                                        |                                                                                                  |
|       | stacked bar plots) might be                                          |                                                                                                  |
| R1.12 | helpful.                                                             | Noted and has been corrected.                                                                    |
| K1.12 | Line 233: Figure 3 refers to area, not climate.                      | Noted and has been corrected.                                                                    |
|       | Tiot climate.                                                        |                                                                                                  |
|       | Line 233: All the others should                                      |                                                                                                  |
|       | point to Figure S5, not Figure S6.                                   |                                                                                                  |
| R1.13 | There are references to regions 1-                                   | We have added a definition of the regions of the                                                 |
|       | 13 in many analyses, but no                                          | Philippines (new Fig. S1) and an explanation of                                                  |
|       | introduction or proper definition of                                 | these regions has been added to section 3.                                                       |
|       | these regions is provided. The first mention appears on line 78,     |                                                                                                  |
|       | without any definition.                                              |                                                                                                  |
| R1.14 | Line 213 claims that the authors                                     | This is a fair comment. The FEH approach is                                                      |
|       | apply the FEH approach                                               | now referenced at the end of section 2, and we                                                   |
|       | described earlier, but there is no                                   | have removed the reference to 'described earlier'                                                |
|       | earlier description. The term is                                     | - it is better to retain the outline of the method                                               |
|       | introduced on line 50, but without sufficient detail.                | here in 4.2 than to explain it earlier and have to repeat key parts here.                        |
|       | Sumoient detail.                                                     | repeat key parts here.                                                                           |
| R1.15 | Improper structure: Lines 220-227                                    | We disagree with this suggestion as the                                                          |
|       | fit better in the data section rather                                | description of variables in 220-7 needs to follow                                                |
|       | than the methods section.                                            | Table 3. The title of section 4.2 is "Predicting high                                            |
|       |                                                                      | magnitude floods from catchment properties" which is entirely appropriate for defining data      |
|       |                                                                      | sources for these variables.                                                                     |
|       |                                                                      |                                                                                                  |
| R1.16 | Line 245: Why are only 71 sites                                      | As it says in line 248 "for 71 sites with at least                                               |
|       | analyzed here instead of the full                                    | 20 annual maxima and for which the GLO                                                           |
|       | 466 sites?                                                           | distribution provided the best fit to the data." We                                              |
|       |                                                                      | have tried to clarify this statement "for the 71 sites that had at least 20"                     |
| R1.17 | The figure captions should be                                        | It is not clear what 'more descriptive of the                                                    |
|       | more descriptive of the settings,                                    | settings' means. The figure captions include very                                                |
|       | but discussions of the                                               | little discussion of the results presented within                                                |
|       | results/patterns should not be                                       | them, and where this is presented we think that it                                               |
|       | included here.                                                       | is valuable to guide the reader's interpretation of                                              |
|       |                                                                      | the figure. Readers may look at figures in isolation and the captions have been prepared to      |
|       |                                                                      | allow figures to be understood without having to                                                 |
|       |                                                                      | read all of the related text.                                                                    |
| R1.18 | Tables could be improved by                                          | The journal format has been followed for the                                                     |
|       | adding delineation lines, clarifying                                 | Tables, although the reviewer's comment about                                                    |
|       | headers (e.g., the last item in the                                  | delineation lines does make sense -some                                                          |
|       | first column of Table 2), and                                        | delineation lines have been added where these                                                    |

|            | removing unnecessary information.                                                                                                                                                                                                                                                                                                                                                                                                                                                                                                                                                                         | are most helpful, and we will rely on the editors' judgement as to whether these are allowable deviations from the journal's formatting guidelines. We have tried to keep tables concise and are unsure what 'unnecessary information' is being referred to here.                                                                                                                                                                                                                                                                                 |
|------------|-----------------------------------------------------------------------------------------------------------------------------------------------------------------------------------------------------------------------------------------------------------------------------------------------------------------------------------------------------------------------------------------------------------------------------------------------------------------------------------------------------------------------------------------------------------------------------------------------------------|---------------------------------------------------------------------------------------------------------------------------------------------------------------------------------------------------------------------------------------------------------------------------------------------------------------------------------------------------------------------------------------------------------------------------------------------------------------------------------------------------------------------------------------------------|
| R1.19      | In summary, I acknowledge that such a study is needed for the selected country, as the authors claim there are no other similar studies to such an extent. While the analyses are comprehensive, the manuscript lacks sufficient clarity in its structure and critical information, which hampers its transferability and overall readability.                                                                                                                                                                                                                                                            | Some of the specific comments made by both reviewers have assisted us in clarifying the paper and we hope that we have met the reviewer's requirements.                                                                                                                                                                                                                                                                                                                                                                                           |
| Reviewer 2 |                                                                                                                                                                                                                                                                                                                                                                                                                                                                                                                                                                                                           |                                                                                                                                                                                                                                                                                                                                                                                                                                                                                                                                                   |
| R2.1       | The paper by Hoey et al., entitled 'Flood estimation for ungauged catchments in the Philippines' aims at delivering design equations to estimate flood magnitudes and frequency in ungauged catchments across the Philippines.  The paper addresses an interesting topic, the design of equations to predict floods in data scarce region. The authors do enormous work to analyse the data. However, the methodology is not well defined/structured.  Overall, the paper lacks clarity and needs to be restructured for better clarity.                                                                  | See previous comments and responses.                                                                                                                                                                                                                                                                                                                                                                                                                                                                                                              |
| R2.2       | The title of the paper does not reflect the content of the paper appropriately. The authors propose and evaluate the index flood approach and multi-variate regression to estimate flood. The approach is evaluated only at catchments with data used to fit the regression equations. In a context of flood estimation in ungauged catchments, the paper is lacking the following steps: 1) a clear definition of the methodology used for regionalization to ungauged catchment, 2) a cross-validation to evaluate the performance of the designed equations in ungauged catchments. It is necessary to | These are all very good suggestions, based on the title of the paper. As noted in our response to Referee 1, this title is not accurate and we have amended the title to fit the contents of the paper. The reviewer's suggestions are exactly what we would have done if the paper retained its original title, and there is merit in presenting such an approach and flow chart to guide readers in the use of the equations. However, that would lengthen the current paper excessively and we consider that this is a separate piece of work. |

|      | evaluate how well the proposed approach would perform in ungauged catchments. Therefore, I suggest the authors consider using some cross-validation approach (i.e., leave-one-out or k-cross validation) where a set of catchments are used to fit the design equations; then the remaining catchment(s) are used as pseudo-ungauged catchment(s) to evaluate the accuracy of the designed equations in ungauged catchments. In addition, the methodology used is not clearly described (see specific comments below). |                                                                                                                                                                                                                                                                  |
|------|------------------------------------------------------------------------------------------------------------------------------------------------------------------------------------------------------------------------------------------------------------------------------------------------------------------------------------------------------------------------------------------------------------------------------------------------------------------------------------------------------------------------|------------------------------------------------------------------------------------------------------------------------------------------------------------------------------------------------------------------------------------------------------------------|
| R2.3 | The paper could benefit from a flow diagram that describes the steps followed to estimate floods at ungauged catchments.                                                                                                                                                                                                                                                                                                                                                                                               | We have added a new table (Table 8) and accompanying description to illustrate application of the method to two sites. We decided not to include a flow diagram as the only requirements for these calculations are measurements of catchment area and rainfall. |
| R2.4 | Specific comments: The authors fit and compare the accuracy of the three distributions showing that the choice of the distribution influences flow estimates and no distribution performs well at all sites. Then, the GLO distribution is selected to predict high flow magnitude in section 4.2 with no justification of this choice.                                                                                                                                                                                | There is an error in line 215, in that the $Q_T$ values used were from the best-fit curve at each site, and not only from the GLO results. This has been corrected.                                                                                       |
| R2.5 | In addition, Equation 5, the factorial standard error for the GLO distribution, is only applicable when number of records is at least 20 years. The study region has only 71 sites (line 245), with more than 20 years data, among the 466 sites retained for the analysis (Line 234). Why choose the GLO distribution if it cannot be applicable to all sites? Why choose only GLO instead of using the best fitted distribution at each site as shown in table 2?                                                    | See previous comment (2.4) – clarifying the curves used will address all of these points.                                                                                                                                                                        |
| R2.6 | •The last paragraph in section 4.1, the authors lists three ways the construct growth curves by combining the curves fitted from each site by region, climate type or catchment areas. Therefore, I                                                                                                                                                                                                                                                                                                                    | The analysis reported in 4.2 is for all catchments individually, and does not used the regionalised growth curves. Section 4.2 does not follow from the final paragraph in section 4.1 as the reviewer assumed.                                                  |

|      | am assuming that the resulting curves are regionalized curves where catchments from within each group(region or climate type, catchment area bin) would be represented by a single curve. Correct In other words, catchments of each group would have a single QT value estimate from the regionalized growth curve. How the author use these results to perform a correlation analysis with catchment properties in Section 4.2 (Line 212: values of QT provided by the GLO analysis previously, were correlated with catchment properties")?                                           | We have revised the wording at the end of 4.1 to make it clear why we are producing combined growth curves and how these can be used, and also edited the start of section 4.2 to clarify that this is concerned with individual catchments.                                                                                                                                                                                                                                                                                                                                                                                                                                                                                                                                                                                                                                                                                                                                              |
|------|------------------------------------------------------------------------------------------------------------------------------------------------------------------------------------------------------------------------------------------------------------------------------------------------------------------------------------------------------------------------------------------------------------------------------------------------------------------------------------------------------------------------------------------------------------------------------------------|-------------------------------------------------------------------------------------------------------------------------------------------------------------------------------------------------------------------------------------------------------------------------------------------------------------------------------------------------------------------------------------------------------------------------------------------------------------------------------------------------------------------------------------------------------------------------------------------------------------------------------------------------------------------------------------------------------------------------------------------------------------------------------------------------------------------------------------------------------------------------------------------------------------------------------------------------------------------------------------------|
| R2.7 | The authors use precipitation dataset covering only 17 years (from 1998-2015) which is not sufficient to derive climatology of the catchments. In addition, this dataset are not available for a period that overlaps with flows records period. As suggestion, the author could use precipitation datasets from reanalysis products (e.g. ERA5) which are available from 1940 to present around the globe. It would cover the same period as flow records for most catchments and provide sufficient long timeseries to derive climatology for precipitation variables used in Table 3. | This is an interesting suggestion. We are aware that none of the available precipitation datasets is perfect for our analysis, as we are combining river flow data from different time periods. We used APHRODITE V2 which, as noted, covers the period 1988-2015. Compared with the other precipitation reanalysis products that are available, APHRODITE has the advantage for our study of being focused on Asia and so is likely to be more reliable in that region. The V2 product is regarded as an improvement over APHRODITE V1 in its representation of extreme events which is another benefit for using it in our study. For both of these reasons, we are satisfied that this data set is appropriate, but note that a comparison between different precipitation estimates would be an interesting area for future work. We note that robust validation of APHRODITE is only just starting (https://doi.org/10.1016/j.jhydrol.2024.132562 preprint published December 2024). |
| R2.8 | Why the fitted curves from individual sites are not used to fit a relationship between flood magnitude and catchment area?                                                                                                                                                                                                                                                                                                                                                                                                                                                               | Table 5 presents the results of these fits and compares them with equations that also include a precipitation term.                                                                                                                                                                                                                                                                                                                                                                                                                                                                                                                                                                                                                                                                                                                                                                                                                                                                       |
| R2.9 | It is unclear how the individual short historical records are combined to generate large dataset that produce consistent results.                                                                                                                                                                                                                                                                                                                                                                                                                                                        | This is covered in section 6.3, and we have tried to make this clearer. In summary, the absence of systematic differences between results from the different data sets (eg Figure 5a-c) provides confidence that we can treat the data from all data sources as a single data set for the analyses presented in the paper.                                                                                                                                                                                                                                                                                                                                                                                                                                                                                                                                                                                                                                                                |
|      | Results presented in Section 5.3.3 (Eq6a-6c) would be better presented in a table with equation and associated R 2 and RMSE in the same way as in Table 5.                                                                                                                                                                                                                                                                                                                                                                                                                    | For consistency within the paper, we have converted equation (6) to table form (Table 6).                                                                                                                                                                                                                                                                                                                                                                                                                                                                                                                                                                                                                                                                                                                                                                                                                                                                                                 |

| R2.10 | Overall, the way the methodology and the results are presented is incoherent and lacks clarity. | Other than the specific comments above, it is difficult to know what the reviewer is referring to here. Both reviewers have made specific comments that will improve the clarity of the results. The mis-leading title of the paper may |
|-------|-------------------------------------------------------------------------------------------------|-----------------------------------------------------------------------------------------------------------------------------------------------------------------------------------------------------------------------------------------|
|       |                                                                                                 | have caused some confusion, and clarifying this will hopefully also aid the clarity of the paper.                                                                                                                                       |

The reviewers' comments have been very valuable in guiding improvements to the manuscript, and we consider that this version is significantly improved as a result of their comments.

Once again, we affirm that this manuscript is original, has not been published previously, and is not under consideration for publication elsewhere. We have no conflicts of interest to disclose. All authors have read and approved the manuscript, and agree to its re-submission

Yours sincerely,

Pamela Louise M. Tolentino

On behalf of the first author, Trevor Hoey and the Catchment Project research team

---

## Referee Report (RR1)

**HESS-2024-188 Review**

This manuscript titled Integrating historical archives and geospatial data to revise flood estimation equations for Philippine rivers is a revision of Flood estimation for ungauged catchments in the Philippines manuscript that has been submitted to Hydrology and Earth System Sciences.

The authors have addressed the comments raised by the reviewers in the previous round of review.

Therefore, it was time consuming to compare the previous and current versions of the manuscript.

The revised manuscript presents an ambitious effort to combine short-duration historical records from diverse sources into a single, coherent analysis for deriving flood design equations at a national scale.

I suggest the authors consider the following points in their revision:

**Major Comments**

Although the authors mentioned uncertainty many times in the manuscript, they did not provide any quantification of the uncertainty in their results. This is a major issue that needs to be addressed.

Also, in general, flood frequency analysis is not a proper method to estimate flood magnitude when you have limited data. Fitting a curve to 7–10 data points is not a reliable method to estimate flood magnitude.

**Abstract**

While the abstract effectively conveys the general research objective and findings, in my opinion it may need some revisions to improve clarity and precision. When you start to mention R² and then express the added value of including the new variables, the sentence is not clear (L25-27). I suggest you revise it. It would benefit from a clearer statement regarding the limitations of the low R² values and the implications for design uncertainty.

**Introduction**

The introduction provides an overview of the study. It briefly describes the importance of catchment area and mean annual rainfall as predictors of flood magnitude. The authors tried to highlight the impact of pooling data from available sources to improve flood estimation when the data are limited in time and space.

I suggest merging the two middle paragraphs of the introduction to make it more concise and clear.

Also, the hypothesis and research questions of the study are not clearly stated in the introduction. It would be better to state them explicitly.

**Methodology**

In the section Data sources, the authors provide a detailed description of the data sources and the process of
data collection. Different sources introduce distinct uncertainties and biases into the analysis. For example, in
Figure 1, some sources (red and blue dots) are more concentrated in regions such as the north of the country,

- while in the west and south—where there is lower rainfall and lower contribution from tropical cyclones—we have no or only one source of data. This may introduce bias in the analysis. The authors should discuss this issue in the manuscript.
- More details regarding the screening criteria for data quality and the rationale behind the selected catchment properties would improve transparency. For example, three sources of data are used in the analysis; while they were recorded differently, in different periods of time, and likely with different measurement techniques, the method of merging these data should be discussed in the manuscript. It is highly likely that the quality of measurements before the 1980s is lower.

**Analysis Methods**

- I am curious to know whether you ever tried to employ two peaks per year or any POT analysis to identify the peaks in the data, instead of only using the annual maximums. This approach would give you more freedom not only to select the highest peak in the year but also the second highest independent peak in the year. This could help you better understand the flood frequency in the region, as the second peak may occur in another season and allow you to better capture your basin's behavior. Then, you could continue to determine Q\_med of the new series of peaks.
- The manuscript provides a thorough description of the curve fitting using L-moments and the subsequent regression analyses. Yet, the discussion on the potential biases arising from combining data of varying quality and the choice of best-fit distributions (with respect to low R² values) deserves further elaboration. Moreover, since the study aims to estimate extreme floods, linear regression may not be the best approach. The authors should consider using a more robust method, such as quantile regression, to account for the non-linear relationship between the predictors and the response variable.
- What is the set threshold of low CvM p-values used to exclude data from the analysis in L183?

**Results**

- The correlation approach in Table 4 does not lead to a new conclusion. The fact that a larger catchment area leads to a higher correlation is not a new finding. It is the same with the DPLBAR variable, the length of the streamflow network, and the mean annual rainfall. Therefore, your addition in Table 5 should be highlighted. I suggest restructuring the results section to emphasize the new findings of the study.
- Perhaps testing and illustrating your approach on only the new dataset as a test case would be a good idea to show the robustness of your approach. This will also help in understanding the uncertainty in the results.
- What would be the expected best R² value by adding the new variables? It would be better to have a benchmark
  to compare the results. What is the ideal R² value for flood frequency analysis in the region? Is the benchmark
  0.92 in Papua New Guinea? You could randomly generate some synthetic data and try to estimate the flood
  frequency analysis to see the ideal R² value.
- As you mentioned, land use change is a major factor in flood frequency analysis, and you employed almost current land use data in the analysis. This is a significant challenge and limitation of the study.
- The abbreviations in this study are not mathematically scientific, such as AREA or RMED. It would be better to use the full names of the variables in the text and use better letters for the variables. For example, A for area, and R m for RMED, and so on.

 Since the results are mainly presented on Q10 and they are not significantly appropriate for flood control and design, it would be better to include a discussion on the results and the limitations of the study.

**Discussion**

- It would be valuable to discuss the limitations (e.g., stationarity assumptions, data quality issues, and land use change) more explicitly and to outline potential paths for future improvement, such as incorporating non-stationary models or enhancing continuous monitoring.
- Tropical cyclones were not part of your investigation; however, they play a role in the discussion.
- Climate change and spatiotemporal variability in the region are not discussed in the manuscript at all, despite the merged data varying over time.
- The discussion section is generally long. I suggest revising it to be brief, more concise, and clear. However, the current form is good for readers to understand the results and limitations of the study.
- The comparison with HEC-HMS modeling lends additional credibility, though the discussion might be expanded to
  explain the practical implications of the observed discrepancies between instantaneous peak flows and daily
  mean flow estimates.

**Conclusion**

• The conclusion is well-structured and effectively summarizes the key findings of the study.

**Minor Comments**

**Abstract:**

- L18: Split the long sentence However, the global ... into two sentences for clarity. The current sentence contains four commas.
- L23: What does national and regional scales mean? Are they two different scales?
- L25: The term GIS-derived is not needed here. You can simply say geospatial catchment characteristics.
- L30: There is a redundancy with the term predictive equation in the same sentence.

**Introduction:**

- L43: The sentence The resulting equations ... is not well connected to the previous sentence. The starting lines are quite good, but there is a gap between the first and second parts of the first paragraph.
- L79: A reference to Figure S1 is needed.
- Please provide a map of the available length of time series in the Philippines. This will help in understanding the
  data availability in the country (L80). Although the time period is indicated in Table 1, it is not clear whether the
  records are continuous or if there are gaps in the data. Alternatively, you can provide some sentences in the text
  to explain this issue.
- How do you define short time series? Is it less than 35 years? (L80). It would be better to provide a definition for short time series or a reference for the definition.

• L84: The FEH abbreviation has already been defined previously in L50.

**Data sources:**

- Figure 1: In the caption, it is mentioned that the four climate types that have been identified for the Philippines (Coronas, 1920). Since the climate types were identified in the 1920s, is there any more recent climate type identification for this region? Given global warming and climate change, the climate types may have changed or been better defined in recent years.
- Figure 1: Please replot panels b and c and use discrete colors instead of gradient colors. Also, Figure 1C does not support any of your results except for a sentence in the conclusion. It would be better to remove it from the manuscript or integrate its insight into your interpretation.
- I suggest moving Table 1 to the supplementary material, as it is not necessary in the main text.

**Analysis Methods:**

- To achieve more consistency in the manuscript text, I suggest adding Q5 and Q50 in Figure 2, and so on, in your text
- Since Table 2 does not show any relation between the size of the catchment, climate type, and the best-fit curve, is there any geographical pattern in the best-fit curve? For example, do catchments in the north of the country have the same best-fit curve? What if you plot the best-fit curve on the map of the country? Usually, subcatchments in the same basin may have the same best-fit curve since they are flow-connected.
- L204: The phrase (Figure S1) show this pattern is unclear. I have not seen this pattern in Figure S1.

  Please revise the text. The mentioned figure is Administrative regions of the Philippines. Since the numbering starts from north to south, it would be better to reorder the legend of the figure to follow the same order instead of alphabetical order.
- L208: The reference to Figure S2 is incorrect. It is currently written as (Figures 2, S1); it should be (Figure S2), as seen in the supplementary material. Also, the figure itself is not well plotted.
- The quality of Figure S3 is too low. It is not readable. Please revise it. The current figure overlays the main curves on top of each other. The area of concentration should be zoomed in to see the differences between the curves on the right part of the x-axis.
- The same applies to Figures S4 and S5. However, they are slightly better. I think these figures can be highlighted for regionalization since they are important for understanding how the curves differ by region, climate type, and catchment size. The current format does not help in understanding the differences between the regions.
- L247: As far as I know, we have free global DEMs with 30 m resolution. So why did you resample the DEM?

**Results:**

- L287: The phrase This contrasts with Meigh's should be moved to the conclusion.
- Table 4: Instead of NA, write in the table.
- Theoretically, your Q\_med should be equivalent to your Q2 when you have a limited length of time series. If you look at Table 4, the columns Q\_med and Q2 are almost identical. Also, the correlation is sensitive to the number of data points.
- Table 5: The alignment of the table is not correct. Please revise it, and make it more readable.
- Figure 5 and subsequent figures: Please elaborate on "normally distributed" in the figure caption, especially for subplots b, c, e, and f.
- Set a fixed significance level for the p-values in the text. In section 5.3.3 it is 0.05, while previously it was 0.01.

• Figure S7 must be revised. The current figure is not readable enough; it is a bit small, and the selected colors do not help the readers understand it. Also, since this figure has three parts, the main body of the manuscript does not support it well.

**Discussion:**

• Figure 8: It seems that the x-axis of panel b is not correct. Please revise it.

This study contributes to hydrological modeling by demonstrating how pooling individually short historical flood records—combined with high-resolution geospatial data—can produce nationally applicable flood estimation equations even in data-sparse tropical regions. The Recommended design equations section is a part that the authors may consider including in their analysis.

---

## Author Response (AR2)

|       | Report 1 – Referee no. 2                   | Response                                                                                              |
|-------|--------------------------------------------|-------------------------------------------------------------------------------------------------------|
| R2.1  | I would like to thank the authors for      | -                                                                                                     |
|       | their efforts to improve the manuscript.   |                                                                                                       |
|       | The paper still lacks clarity regarding    |                                                                                                       |
|       | methodology and the did not address        |                                                                                                       |
|       | comments raised. See below:                |                                                                                                       |
| R2.2  | Although the authors claim that they       | Lines 170-180 explain how the three distributions                                                     |
|       | have used the best-fit curve at each site, | were fitted to each individual site, and the highest                                                  |
|       | their description of methodology and       | CvM p-value was then used to select the                                                               |
|       | presented results show the GLO was         | appropriate distribution for further analysis. Line                                                   |
|       | used for all sites. The last paragraph in  | 227 corrected to 'best-fit curves' and additional                                                     |
|       | section 4.1 and Figure 4 caption clearly   | text on line 180 to remove doubt.                                                                     |
|       | states that GLO was used for all sites.    | Figure 4 continued and the total that CLO con-                                                        |
|       |                                            | Figure 4 caption does not state that GLO was                                                          |
|       |                                            | used for all sites, rather 'GLO curves for data                                                |
|       |                                            | pooled from all sites', 'from bins of                                                                 |
|       |                                            | catchment area ' etc. which is quite different from saying that GLO was used for all sites. On |
|       |                                            | figure 4, only 4(a) shows data from single sites,                                                     |
|       |                                            | and these include GLO, Weibull and LP-III fits. We                                                    |
|       |                                            | have edited the Figure 4 caption to try to avoid                                                      |
|       |                                            | any confusion.                                                                                        |
| R2.3  | The methodology section lacks clarity.     | It is difficult to know what the reviewer is looking                                                  |
|       | For now it only describes the              | for here. The analysis includes linear and multiple                                                   |
|       | methodology to fit the curves and          | regressions that are introduced where                                                                 |
|       | predicting high magnitude floods from      | appropriate along with notes of, for example,                                                         |
|       | catchment properties. The methodology      | transformations applied to variables. Adding a                                                        |
|       | section should clearly describe the steps  | section to the end of section 4 that says that                                                        |
|       | the authors have taken to perform their    | these equations will be used would add length,                                                        |
|       | analysis and evaluation of the results.    | introduce repetition and would not help readers.                                                      |
|       |                                            | We contend that regression analysis is sufficiently                                                   |
|       |                                            | routine not to require a primer in the                                                                |
|       |                                            | methodology section. We have, however, moved                                                          |
|       |                                            | section 5.3.1 to 4.2, agreeing with the reviewer                                                      |
|       |                                            | that this is a more appropriate location for this                                                     |
|       |                                            | descriptive information.                                                                              |
| R2.4  | What is the purpose of showing fse if it   | Although not applicable to the majority of sites,                                                     |
|       | not applicable to the majority of sites?   | fse is the only way to understand the magnitude                                                       |
|       |                                            | of errors within the data set. Having some error                                                      |
|       |                                            | estimates provides an indication of the likely                                                        |
|       |                                            | magnitude of errors at all sites, although we appreciate that errors are greater for sites with       |
|       |                                            | shorter records. An additional sentence has been                                                      |
|       |                                            | added at the end of section 5.1 to confirm this.                                                      |
| R2.5  | Caption of figures and tables are too      | We have followed the journal's style guides and                                                       |
| 114.5 | long with unnecessary information that     | previous practice. Specifically, we have used                                                         |
|       | readers can easily understand by           | comprehensive captions to ensure accessibility of                                                     |
|       | themselves. For instance, there is no      | the paper to readers with visual impairment who                                                       |
|       | need to describe what colors and           | may be reliant on text translation. This also                                                         |
|       | symbols mean if the figure has a           | benefits readers, most commonly in the Global                                                         |
|       | legend.                                    | South, who have slow internet connections that                                                        |
|       |                                            |                                                                                                       |

| R2.6 | There is no need to show number of excluded sites with records of at least 7 years in Table 2. It has been clearly made at this stage that they are excluded and analysis considers only 513 sites. | This is true, but for the avoidance of doubt and to enable Table 2 to be used in isolation we prefer to retain this information in the table.                                                                                                                                                                                               |
|------|-----------------------------------------------------------------------------------------------------------------------------------------------------------------------------------------------------|---------------------------------------------------------------------------------------------------------------------------------------------------------------------------------------------------------------------------------------------------------------------------------------------------------------------------------------------|
| R2.7 | Line 205- 209. The ratios between flow estimated from different curves could be better presented as a Table.                                                                                        | We have thought about this issue at some length. The paper includes 8 tables already and we are reluctant to add further length by including another one. The information in the text is dense but can be followed easily by interested readers. Those readers who are less concerned with these details can move on to the next paragraph. |
| R2.8 | Table 4 is better fitted in section 4.2.                                                                                                                                                            | Agreed and actioned – see previous comment.                                                                                                                                                                                                                                                                                                 |

|      | Report 2 – Referee no. 3                                                                                                                                                                                            | Response                                                                                                                                                                                                                                                                                                                                                                                                                                                                                                                                                                                                                                                                                                                                                                                                                                                               |
|------|---------------------------------------------------------------------------------------------------------------------------------------------------------------------------------------------------------------------|------------------------------------------------------------------------------------------------------------------------------------------------------------------------------------------------------------------------------------------------------------------------------------------------------------------------------------------------------------------------------------------------------------------------------------------------------------------------------------------------------------------------------------------------------------------------------------------------------------------------------------------------------------------------------------------------------------------------------------------------------------------------------------------------------------------------------------------------------------------------|
|      | Major Comments                                                                                                                                                                                                      |                                                                                                                                                                                                                                                                                                                                                                                                                                                                                                                                                                                                                                                                                                                                                                                                                                                                        |
| R3.1 | Although the authors mentioned uncertainty many times in the manuscript, they did not provide any quantification of the uncertainty in their results. This is a major issue that needs to be addressed.             | We strongly disagree with this comment. There are many places where uncertainty is considered and demonstrated – for example residuals plots in Figures 5 and 6, residuals mapping in Figure 7, R 2 and se information in Tables 6,7, and related discussion in the text, and the comparative results in section 6.2.2.                                                                                                                                                                                                                                                                                                                                                                                                                                                                                                                                     |
| R3.2 | Also, in general, flood frequency analysis is not a proper method to estimate flood magnitude when you have limited data. Fitting a curve to 7–10 data points is not a reliable method to estimate flood magnitude. | The paper makes clear that the Philippines is a data poor environment and that we are trying to produce usable equations for flood magnitude estimation when data are sparse and will continue to be sparse. The paper makes clear that our approach is to use all available data and to pool this to maximise the information that we can derive from what is available, while being realistic about remaining uncertainties. This is not an uncommon approach, although in more data-rich settings there are better methods available as we note – see lines 55-61, and 62-71 for Philippines-specific commentary. We attempt in the final paragraph of the paper to assess the way forward for flood estimation in the Philippines, and other data-poor settings and note that a large majority of the global population reside in data-poor countries and regions. |
| R3.3 | Abstract                                                                                                                                                                                                            | This is a very helpful comment on the abstract, and we have re-written the                                                                                                                                                                                                                                                                                                                                                                                                                                                                                                                                                                                                                                                                                                                                                                                             |

While the abstract effectively conveys the general research objective and findings, in my opinion it may need some revisions to improve clarity and precision. When you start to mention R2 and then express the added value of including the new variables, the sentence is not clear (L25-27). I suggest you revise it. It would benefit from a clearer statement regarding the limitations of the low R2 values and the implications for design uncertainty.

abstract to improve its clarity and to remove some potential areas of misunderstanding.

**R3.4 Introduction**

The introduction provides an overview of the study. It briefly describes the importance of catchment area and mean annual rainfall as predictors of flood magnitude. The authors tried to highlight the impact of pooling data from available sources to improve flood estimation when the data are limited in time and space.

I suggest merging the two middle paragraphs of the introduction to make it more concise and clear.

Also, the hypothesis and research questions of the study are not clearly stated in the introduction. It would be better to state them explicitly.

We have decided not to merge paragraphs as the current structure appears, to us, to separate distinct aspects of the background to the study.

We note the comment regarding research questions and have amended the final paragraph of the introduction accordingly.

**R3.5 **Methodology**

In the section `Data sources`, the authors provide a detailed description of the data sources and the process of data collection. Different sources introduce distinct uncertainties and biases into the analysis. For example, in Figure 1, some sources (red and blue dots) are more concentrated in regions such as the north of the country, while in the west and south—where there is lower rainfall and lower contribution from tropical cyclones—we have no or only one source of data. This may introduce bias in the analysis. The authors should discuss this issue in the manuscript.

This is a valid comment, but we have to work with the available data. The maps of residuals in the paper do not show systematic regional patterns, nor do residuals from regressions show systematic effects from different data sources (Figure 5 a-c; Figure S5c). These results give us confidence in the comparability of the data from different sources, and we have checked the methodology used in collecting the data and it is consistent (stagedischarge curves, often with contextual notes about issues regarding reliability of measurement of the highest flows). In section 6.3, we discuss if there are any issues surrounding the data and their amalgamation. One additional sentence "Comparison of results from different data sources (e.g. Figure 5(a-c)) shows no statistically significant differences between results from analysis for each of the data sets, so supporting our

amalgamation of the data from different sources for aggregated analysis." has been added to 6.3 to re-state our confidence in the data sources. R3.6 We think that the paper contains More details regarding the screening criteria sufficient information regarding the for data quality and the rationale behind the data selection, and also refer to the selected catchment properties would previous comment and response. improve transparency. For example, three sources of data are used in the analysis; while they were recorded differently, in Although measurement methods were different periods of time, and likely with different, the principles of stagedifferent measurement techniques, the discharge gauging have been largely method of merging these data should be unchanged for two centuries and we discussed in the manuscript. It is highly likely have confidence in all of the data. The that the quality of measurements before the books containing data for the 1910-1980s is lower. 1920s include excellent and detailed information on rating curves and site characteristics. For example, we have been able to use rating data to infer periods of river bed aggradation or degradation at some locations. Hence, the quality of measurements is likely to have been higher for the earlier data, although we do not have rating curves from more recent data collection periods to assess this further. Note that the analysis here uses only annual maxima, rather than full flow records which would be more affected by gaps in the data record and other quality issues. R3.7 **Analysis Methods** This is an interesting suggestion. It would have been possible, although I am curious to know whether you ever tried some of the flow records do not contain to employ two peaks per year or any POT sufficient information to identify all POT analysis to identify the peaks in the data, events (for example, at some sites we instead of only using the annual maximums. only have peak flow available for each This approach would give you more freedom month of the year and at others only not only to select the highest peak in the the annual maxima are provided). year but also the second highest independent peak in the year. This could Note also that in this tropical help you better understand the flood environment there is strong and frequency in the region, as the second peak generally consistent flow seasonality. may occur in another season and allow you to better capture your basin's behavior. If the paper is accepted, all of our data Then, you could continue to determine will be openly available for others to use Q\_med of the new series of peaks. and to undertake additional analyses.

| R3.8  | The manuscript provides a thorough description of the curve fitting using L-moments and the subsequent regression analyses. Yet, the discussion on the potential biases arising from combining data of varying quality and the choice of best-fit distributions (with respect to low R² values) deserves further elaboration. Moreover, since the study aims to estimate extreme floods, linear regression may not be the best approach. The authors should consider using a more robust method, such as quantile regression, to account for the non-linear relationship between the predictors and the | There are several issues raised here, and we concur with the overall premise of needing to use the most appropriate methods for the data that are available. As noted above, we have had to combine data from different sources and to rely on relatively short records in order to produce a data set with national coverage. Quantile regression would have required longer and more complete data records than are available for nearly all of our sites, so severely limiting our analysis. We have previously applied quantile regression |
|-------|---------------------------------------------------------------------------------------------------------------------------------------------------------------------------------------------------------------------------------------------------------------------------------------------------------------------------------------------------------------------------------------------------------------------------------------------------------------------------------------------------------------------------------------------------------------------------------------------------------|------------------------------------------------------------------------------------------------------------------------------------------------------------------------------------------------------------------------------------------------------------------------------------------------------------------------------------------------------------------------------------------------------------------------------------------------------------------------------------------------------------------------------------------------|
|       | response variable.                                                                                                                                                                                                                                                                                                                                                                                                                                                                                                                                                                                      | methods (Franco-Villoria et al., 2019, DOI: 10.1002/env.2522) and appreciate the potential of this approach where suitable data are available.                                                                                                                                                                                                                                                                                                                                                                                                 |
|       |                                                                                                                                                                                                                                                                                                                                                                                                                                                                                                                                                                                                         | Further, the analysis methods used are standard (eg Kjeldsen, 2013) and we adopt this methodology to ensure consistency with previous work. The paper comments on some of the background analysis that we undertook to assess the data (Figure S6 shows cross-correlations that show the nature of relationships between all variables utilised).                                                                                                                                                                                              |
|       |                                                                                                                                                                                                                                                                                                                                                                                                                                                                                                                                                                                                         | We are considering undertaking further analysis of these data, potentially using GAM methods, but consider this to be a separate project from the current work. The potential value of design equations that use established methods in a datasparse country should not be undervalued, and we consider our approach to be the most robust and reliable way to develop these equations at this time.                                                                                                                                           |
| R3.9  | What is the set threshold of low CvM p-values used to exclude data from the analysis in L183?                                                                                                                                                                                                                                                                                                                                                                                                                                                                                                           | It is best not to interpret CvM p-values against a critical alpha value, but to compare the CvM statistics between distributions (Asquith, 2020). The median p-value of best-fit curves was 0.93 and this has been noted in the text.                                                                                                                                                                                                                                                                                                          |
| R3.10 | Results - The correlation approach in Table 4 does                                                                                                                                                                                                                                                                                                                                                                                                                                                                                                                                                      | This comment is understood and has
been addressed in response to other
reviewer's comments by moving Table 4                                                                                                                                                                                                                                                                                                                                                                                                                             |
|       | not lead to a new conclusion. The fact that a larger catchment area leads to a higher                                                                                                                                                                                                                                                                                                                                                                                                                                                                                                                   | into section 4.2 where it is presented as background information.                                                                                                                                                                                                                                                                                                                                                                                                                                                                              |

| R3.11 | correlation is not a new finding. It is the same with the DPLBAR variable, the length of the streamflow network, and the mean annual rainfall. Therefore, your addition in Table 5 should be highlighted. I suggest restructuring the results section to emphasize the new findings of the study.  Perhaps testing and illustrating your                                                                               | We would agree with this suggestion IF                                                                                                                                                                                                                                                                                                                                                        |
|-------|------------------------------------------------------------------------------------------------------------------------------------------------------------------------------------------------------------------------------------------------------------------------------------------------------------------------------------------------------------------------------------------------------------------------|-----------------------------------------------------------------------------------------------------------------------------------------------------------------------------------------------------------------------------------------------------------------------------------------------------------------------------------------------------------------------------------------------|
|       | approach on only the new dataset as a test case would be a good idea to show the robustness of your approach. This will also help in understanding the uncertainty in the results.                                                                                                                                                                                                                                     | the aim of the paper was to test a new method. However, our aim is to produce reliable and robust design equations for the Philippines and so this calibrate and test approach is less appropriate.  We note that several of the plots differentiate the different datasets. If there was bias related to the data sources, this would be apparent in these plots.                            |
| R3.12 | What would be the expected best R 2 value by adding the new variables? It would be better to have a benchmark to compare the results. What is the ideal R 2 value for flood frequency analysis in the region? Is the benchmark 0.92 in Papua New Guinea? You could randomly generate some synthetic data and try to estimate the flood frequency analysis to see the ideal R 2 value. | Previous global analysis (eg Meigh et al., 1997) has reported R 2 values from 0.61 (Kerala) to 0.92 (PNG). Equations that go beyond catchment area and one rainfall variable can improve R 2 values slightly (eg in Indonesia improvement from 0.881 to 0.889 by adding both catchment slope and lake area terms).                                                      |
| R3.13 | As you mentioned, land use change is a major factor in flood frequency analysis, and you employed almost current land use data in the analysis. This is a significant challenge and limitation of the study.                                                                                                                                                                                                           | Yes, we acknowledge this and make some comments to this effect in the paper. At the catchment scale, the influence of changes over time may be less than at smaller scales. Historical land-use data do not exist for the Philippines, but we note that none of the catchments in the study is extensively urbanised. Similar challenges would be encountered in almost any tropical country. |
| R3.14 | The abbreviations in this study are not mathematically scientific, such as AREA or RMED. It would be better to use the full names of the variables in the text and use better letters for the variables. For example, A for area, and R_m for RMED, and so on.                                                                                                                                                         | We have followed convention from the UK Flood Estimation Handbook (FEH) in naming variables and consider that it is appropriate to retain these variable names to facilitate easy comparison with a wide range of previous studies.                                                                                                                                                           |
| R3.15 | Since the results are mainly presented on Q10 and they are not significantly appropriate for flood control and design, it would be better to include a discussion on the results and the limitations of the study.                                                                                                                                                                                                     | We note Q2 (close to the geomorphologically effective bankfull flood) and Q10 results in the tables in the paper. Q10 was selected for presentation rather than Q100 as the relatively short time series available for                                                                                                                                                                        |

|       |                                                                                                                                                                                                                                                                                     | analysis make estimation of Q100 less reliable.  We have added a note to the conclusions to make the point about the limitation of Q10 for design purposes.                                                                                                                                                                                                                                                                              |
|-------|-------------------------------------------------------------------------------------------------------------------------------------------------------------------------------------------------------------------------------------------------------------------------------------|------------------------------------------------------------------------------------------------------------------------------------------------------------------------------------------------------------------------------------------------------------------------------------------------------------------------------------------------------------------------------------------------------------------------------------------|
| R3.16 | - It would be valuable to discuss the limitations (e.g., stationarity assumptions, data quality issues, and land use change) more explicitly and to outline potential paths for future improvement, such as incorporating non-stationary models or enhancing continuous monitoring. | Section 6.4 does address the limitations of the study – stationarity and alternative approaches are referenced in the final paragraph of 6.4. Section 6.4 also contains suggestions for grouping of catchments for analysis, noting that this may not involve grouping adjacent catchments. We consider that there is sufficient acknowledgment and discussion of limitations and potential enhancements to the study through the paper. |
| R3.17 | Tropical cyclones were not part of your investigation; however, they play a role in the discussion.                                                                                                                                                                                 | Rainfall contributions from cyclones are introduced in Figure 1, and they are mentioned in the discussion as an issue that may require further consideration to account for possible changes to precipitation patterns due to climate change. A sentence has been added to the end of section 6.4 for clarity.                                                                                                                           |
| R3.18 | Climate change and spatiotemporal variability in the region are not discussed in the manuscript at all, despite the merged data varying over time.                                                                                                                                  | Within the limitations of what is an already lengthy manuscript, we do report the most recent (Tolentino et al., 2016) assessment of future hydroclimatic change. There are few studies of climate change in the Philippines over the past century and a lack of data to make reliable statements regarding past changes.                                                                                                                |
| R3.19 | The discussion section is generally long. I suggest revising it to be brief, more concise, and clear. However, the current form is good for readers to understand the results and limitations of the study.                                                                         | This comment is somewhat contradictory. Several of this reviewer's comments ask for more discussion of the context and limitations in the study, so it is difficult to see how we could shorten the discussion without oversimplification. Some minor changes have been made in response to this, and the other, reviewer's suggestions that we hope clarify our reasoning.                                                              |
| R3.20 | The comparison with HEC-HMS modeling lends additional credibility, though the discussion might be expanded to explain the practical implications of the observed                                                                                                                    | We agree that we could expand on this further, but note the reviewer's previous comment suggesting making the discussion shorter. We have tried to use the HEC-HMS comparison fairly, and                                                                                                                                                                                                                                                |

|       | discrepancies between instantaneous peak          | not to over-state its value as the HEC- |
|-------|---------------------------------------------------|-----------------------------------------|
|       | flows and daily mean flow estimates.              | HMS modelling relied on several         |
|       | ,                                                 | assumptions that we are not able to     |
|       |                                                   | evaluate.                               |
| R3.21 | Conclusion                                        | Noted and appreciated.                  |
|       | The conclusion is well-structured and             |                                         |
|       | effectively summarizes the key findings of        |                                         |
|       | the study.                                        |                                         |
|       | Minor comments                                    |                                         |
| R3.22 | Abstract:                                         | Edited to improve clarity.              |
| NO.LL | 7 isstract.                                       | Lanca to improve clarity.               |
|       | L18: Split the long sentence `However, the        |                                         |
|       | global` into two sentences for clarity. The       |                                         |
|       | current sentence contains four commas.            |                                         |
| R3.23 | L23: What does `national and regional scales`     | Abstract clarified 'both national and   |
| NO.LO | mean? Are they two different scales?              | regional'. Section 3 explains the basis |
|       | mean. The they two amerene scales.                | for regionalization.                    |
| R3.24 | L25: The term `GIS-derived` is not needed         | Done                                    |
| 1.5.2 | here. You can simply say `geospatial              |                                         |
|       | catchment characteristics`.                       |                                         |
| R3.25 | L30: There is a redundancy with the term          | Done                                    |
| NS.ES | `predictive equation` in the same sentence.       | Bone                                    |
| R3.26 | Introduction:                                     | A connecting sentence has been added:   |
|       |                                                   | "Understanding flood magnitude and      |
|       | L43: The sentence `The resulting                  | frequency is crucial for designing      |
|       | equations` is not well connected to the           | mitigation strategies, and this         |
|       | previous sentence. The starting lines are         | understanding relies on using           |
|       | quite good, but there is a gap between the        | empirical analyses to generate          |
|       | first and second parts of the first paragraph.    | predictive models."                     |
|       |                                                   |                                         |
| R3.27 | L79: A reference to Figure S1 is needed.          | Done                                    |
| R3.28 | Please provide a map of the available length      | This is quite a complex issue – the     |
|       | of time series in the Philippines. This will help | opening paragraph of section 3 does     |
|       | in understanding the data availability in the     | address the nature of the records and   |
|       | country (L80). Although the time period is        | explains how we have combined some      |
|       | indicated in Table 1, it is not clear whether     | records from adjacent measuring sites.  |
|       | the records are continuous or if there are        |                                         |
|       | gaps in the data. Alternatively, you can          |                                         |
|       | provide some sentences in the text to             |                                         |
|       | explain this issue.                               |                                         |
| R3.29 | How do you define short time series? Is it        | We have changed to 3-20 years, as 20    |
|       | less than 35 years? (L80). It would be better     | years is often used as an arbitrary     |
|       | to provide a definition for short time series     | threshold for undertaking flood         |
|       | or a reference for the definition.                | frequency analysis. 'Long' has been     |
|       |                                                   | replaced by 'multi-decadal' where the   |
|       |                                                   | length of data records is first         |
|       |                                                   | introduced.                             |
| R3.30 | L84: The `FEH` abbreviation has already been      | This has been corrected.                |
|       | defined previously in L50.                        |                                         |

| R3.31 | Data sources:                                                                                                                                                                                                                                                                                                                                                                                                                           | The Coronas (1920) classification                                                                                                                                                                                                                                                                                                                                                                                                                              |
|-------|-----------------------------------------------------------------------------------------------------------------------------------------------------------------------------------------------------------------------------------------------------------------------------------------------------------------------------------------------------------------------------------------------------------------------------------------|----------------------------------------------------------------------------------------------------------------------------------------------------------------------------------------------------------------------------------------------------------------------------------------------------------------------------------------------------------------------------------------------------------------------------------------------------------------|
|       | Figure 1: In the caption, it is mentioned that `the four climate types that have been identified for the Philippines (Coronas, 1920)`. Since the climate types were identified in the 1920s, is there any more recent climate type identification for this region? Given global warming and climate change, the climate types may have changed or been better defined in recent years.                                                  | continues to be used for the Philippines, and is extensively referenced in climate and hydrological publications. We are unaware of more recent re-evaluations of these climate types, and the familiarity of Coronas' classification to Philippines readers will aid their understanding and ability to interpret our results.                                                                                                                                |
|       | Figure 1: Please replot panels b and c and use discrete colors instead of gradient colors. Also, Figure 1C does not support any                                                                                                                                                                                                                                                                                                         | We do not agree with the recommended re-plotting of 1b and 1c, as gradient colours better reflect the                                                                                                                                                                                                                                                                                                                                                          |
|       | of your results except for a sentence in the conclusion. It would be better to remove it from the manuscript or integrate its insight into your interpretation.                                                                                                                                                                                                                                                                         | interpolation that has been undertaken to generate the maps. Figure 1c provides context here – as noted, it is referred to in the discussion. Had we omitted Fig 1c, we would have expected reviewers to ask about the importance of tropical cyclones!                                                                                                                                                                                                        |
| R3.31 | I suggest moving Table 1 to the supplementary material, as it is not necessary in the main text.                                                                                                                                                                                                                                                                                                                                        | We are following journal guidelines that discourage long supplementary materials. As the paper focuses on integrating data sets, we consider that it is useful for readers to see these data sets described at the outset.                                                                                                                                                                                                                                     |
| R3.32 | Analysis Methods:  To achieve more consistency in the manuscript text, I suggest adding Q5 and Q50 in Figure 2, and so on, in your text.                                                                                                                                                                                                                                                                                                | This could be done, but would add length and complexity to the paper. The full data set will be made available if the paper is published enabling users to compute Q5, Q20, Q50 or other return                                                                                                                                                                                                                                                                |
| R3.33 | Since Table 2 does not show any relation between the size of the catchment, climate type, and the best-fit curve, is there any geographical pattern in the best-fit curve? For example, do catchments in the north of the country have the same best-fit curve? What if you plot the best-fit curve on the map of the country? Usually, subcatchments in the same basin may have the same best-fit curve since they are flow-connected. | periods as they wish.  This is an interesting issue that we looked at in detail when producing the plots for the paper. There are few consistent patterns, but we do observe some consistency within large catchments as suggested. Given the complexities introduced by variable lengths of record, a map of best-fit curve type is not especially helpful for this paper and would require extensive explanation to make sense of what is a complex pattern. |
| R3.34 | L204: The phrase `(Figure S1) show this pattern` is unclear. I have not seen this pattern in Figure S1. Please revise the text. The mentioned figure is `Administrative regions of the Philippines`. Since the numbering starts from north to south, it would be better to reorder the legend of the                                                                                                                                    | We have revised Figure S1 to rearrange the numbering in the legend and rechecked the text to correct reference of figures.                                                                                                                                                                                                                                                                                                                                     |

|       | figure to follow the same order instead of        |                                             |
|-------|---------------------------------------------------|---------------------------------------------|
|       | alphabetical order.                               |                                             |
| R3.35 | L208: The reference to Figure S2 is incorrect.    | We have corrected the numbering of          |
|       | It is currently written as `(Figures 2, S1)`; it  | the Figures.                                |
|       | should be `(Figure S2)`, as seen in the           |                                             |
|       | supplementary material. Also, the figure          |                                             |
|       | itself is not well plotted.                       |                                             |
| R3.36 | The quality of Figure S3 is too low. It is not    | The symbol size on Figure 3 has been        |
|       | readable. Please revise it. The current figure    | reduced to aid clarity. The high-           |
|       | overlays the main curves on top of each           | resolution figures that will be used for    |
|       | other. The area of concentration should be        | final production are clear enough on-       |
|       | zoomed in to see the differences between          | screen.                                     |
|       | the curves on the right part of the x-axis.       |                                             |
| R3.37 | The same applies to Figures S4 and S5.            | See previous comment.                       |
|       | However, they are slightly better. I think        |                                             |
|       | these figures can be highlighted for              | We have examined plots on a region-         |
|       | regionalization since they are important for      | by-region basis and no patterns emerge      |
|       | understanding how the curves differ by            | from this. As the results in the paper      |
|       | region, climate type, and catchment size. The     | show, the regions are not hydrologically    |
|       | current format does not help in                   | distinct. As these regions are very         |
|       | understanding the differences between the         | widely used in the Philippines, as they     |
|       | regions.                                          | are administrative regions, it is useful to |
|       |                                                   | include some regionalisation in the         |
|       |                                                   | paper for the benefit of local users.       |
|       |                                                   | However, climate type and catchment         |
|       |                                                   | area are more informative as ways of        |
|       |                                                   | grouping the data than the regions.         |
| R3.38 | L247: As far as I know, we have free global       | It is better to resample a higher           |
|       | DEMs with 30 m resolution. So why did you         | resolution DEM than to use data             |
|       | resample the DEM?                                 | collected at lower resolution.              |
| R3.39 | Results:                                          | We have not done this as this sentence      |
|       |                                                   | only makes sense within its current         |
|       | L287: The phrase `This contrasts with             | context – moving it to the conclusion       |
|       | Meigh's` should be moved to the conclusion.       | would require 2 or 3 sentences of           |
|       | 3                                                 | explanation that would be out of place      |
|       |                                                   | in the conclusion.                          |
| R3.40 | Table 4: Instead of `NA`, write `-` in the table. | We have explained NA in the caption to      |
|       |                                                   | Table 4.                                    |
| R3.41 | Theoretically, your Q_med should be               | Noted and agreed.                           |
|       | equivalent to your Q2 when you have a             |                                             |
|       | limited length of time series. If you look at     |                                             |
|       | Table 4, the columns Q_med and Q2 are             |                                             |
|       | almost identical. Also, the correlation is        |                                             |
|       | sensitive to the number of data points.           |                                             |
|       | Table 5: The alignment of the table is not        | Done (partly) and will require final        |
| R3.42 | correct. Please revise it, and make it more       | alignment by the journal in                 |
| R3.42 | 12 USB USB                                        |                                             |
| R3.42 | readable                                          | production                                  |
| R3.42 | readable.                                         | production                                  |
|       |                                                   |                                             |
| R3.42 | Figure 5 and subsequent figures: Please           | Have added (Gaussian) in Figure 5           |
|       |                                                   |                                             |

| R3.44 | Set a fixed significance level for the p-values in the text. In section 5.3.3 it is 0.05, while previously it was 0.01.                                                                                                                                                                                                                                                                                                                                                                                      | We do not use a significance level to assess our calculated p-values against; rather, all of the p-values reported are direct computations of the probabilities of Type I errors. This provides readers with a ready way of assessing significance, rather than setting arbitrary pass/fail significance levels.                                                                                                                                                                    |
|-------|--------------------------------------------------------------------------------------------------------------------------------------------------------------------------------------------------------------------------------------------------------------------------------------------------------------------------------------------------------------------------------------------------------------------------------------------------------------------------------------------------------------|-------------------------------------------------------------------------------------------------------------------------------------------------------------------------------------------------------------------------------------------------------------------------------------------------------------------------------------------------------------------------------------------------------------------------------------------------------------------------------------|
| R3.45 | Figure S7 must be revised. The current figure is not readable enough; it is a bit small, and the selected colors do not help the readers understand it. Also, since this figure has three parts, the main body of the manuscript does not support it well.                                                                                                                                                                                                                                                   | Some of the lines will be enhanced for the final production version.  The colours have been chosen for consistency with other figures in the paper (which in turn are based on previous literature). We have tested the figures using the Coblis colourblindness simulator and have selected combinations of colours and line styles that enhance the accessibility of the figures. Note that on-screen high resolution figures as will be used in final production are very clear. |
| R3.46 | Discussion: Figure 8: It seems that the x-axis of panel b is not correct. Please revise it.                                                                                                                                                                                                                                                                                                                                                                                                           | Figure 8(b) x-axis label is correct. This plot is checking for a catchment size effect (and bias) in the results.                                                                                                                                                                                                                                                                                                                                                                   |
| R3.47 | This study contributes to hydrological modeling by demonstrating how pooling individually short historical flood records—combined with high-resolution geospatial data—can produce nationally applicable flood estimation equations even in datasparse tropical regions. The `Recommended design equations` section is a part that the authors may consider including in their analysis.\ I suggest authors consider the above points in their revision and I look forward to seeing the revised manuscript. | Noted, and Tables 6-8 provide design equations and an example of their use.                                                                                                                                                                                                                                                                                                                                                                                                         |

---

## Author Response (AR3)

**14 August 2025**

**Dr Rohini Kumar**

Handling Editor Hydrology and Earth Systems Sciences

Reference: HESS-2024-188

Dear Editor,

Thank you for handling our manuscript and for reopening the review process.

We appreciate Referee #3's thoughtful comments and are broadly in agreement with their assessment of the manuscript. However, undertaking all of the suggested additional analyses, particularly those involving alternative methodologies, would substantially alter the paper's focus and significantly increase its length. To address these comments within the current scope, we have taken two key actions:

- 1. Addition of a new paragraph in the discussion in 6.1.2 which addresses some of the specific comments from Referee #3, and a new Figure S10 which presents both confidence and prediction intervals for our recommended design equations. We considered adding Figure S10 the main manuscript but have decided that this would add length, but it will be readily available to readers who wish to utilise this information. The data are openly available, and other researchers are welcome to undertake analyses to build upon our contribution.
- 2. Inclusion of some additional commentary in section 6.4 to further specify potential alternative analytical approaches and directions for extending this work. Some further minor edits have been incorporated elsewhere in the paper to emphasise these issues.

Following the comments of Referee #3, we have updated the manuscript in several places. Details are in Table 1 which includes Referee #3's comments and suggestions. In addition, referring back to previous referees' comments, we have undertaken the following:

- Some minor editing throughout the manuscript to ensure clarity of meaning and to correct some minor typos; note that Figure 6 has been revised due to incorrect axis limits on the previous version which led to some data being omitted from the figure
- The notation A is used for catchment area mostly rather than AREA (previous reviewers comment on notation), but AREA is explained for consistency with the FEH methodology
- Table 2 rows re-ordered to emphasise final choice of 466 sites, which has been a source of continued confusion in earlier reviews

**Table 1**. Referee #3 comments and responses

| Comment                                                  | Response         |
|----------------------------------------------------------|------------------|
| Uncertainty Quantification                               | Noted and agreed |
| The manuscript presents residual plots                   |                  |
| (Figures 5 and 6), residual mapping (Figure              |                  |
| 7), and summary statistics, including R 2 and |                  |
| standard error (Tables 6 and 7), as                      |                  |
| indicators of uncertainty. While informative             |                  |
| for model diagnostics, these are not                     |                  |
| substitutes for a formal uncertainty analysis.           |                  |

A robust treatment of uncertainty in flood design requires:

- Confidence or prediction intervals for Q estimates (e.g., Q2, Q10, Q100)
- Sensitivity analysis to evaluate how changes in AREA, RMED, and other predictors affect outcomes
- Uncertainty propagation to assess how uncertainties in input data, model parameters, and structural assumptions influence the final flood estimates across regions

We have included a commentary and new figure S10 as described above.

Further sensitivity and uncertainty analyses have not been incorporated due to the considerable additional length that these would add to the paper. We have noted that multi-variate sensitivity analysis will be a useful way of extending our analysis, although with the caveat that the reliability of some of the input variables remains uncertain (as already noted in previous versions of the manuscript).

**For instance:**

Dashed lines in Figure 5 are visual guides (1:1, 1:2, 2:1) and do not represent statistical intervals.

Figure 8 includes approximate 95% prediction intervals (±2σ) for selected points, but these are only applied to the DREAM model comparison and not to the core regression framework.

core regression framework.

While steps such as data screening, exclusion of unreliable records, and residual inspection show good methodological care, they do not quantify how uncertainty in inputs translates into uncertainty in outputs. Without prediction intervals or sensitivity diagnostics, practitioners cannot assess the confidence of the resulting design estimates—an essential requirement for engineering or policy use.

These observations are all correct. Note that Figure 8 only shows confidence intervals for selected points to maintain clarity on the plot, and these intervals apply to all points.

To avoid lengthening the paper, additional figures have not been included. However, Figure S10 has been added which shows the regression equations from Table 5 with 95% confidence intervals. Comments have been added to the text to direct readers to this figure.

**Overall Assessment**

This study addresses a significant problem in a complex data environment and makes a valuable contribution to the field. However, the lack of quantitative uncertainty analysis and the quality of the figures restrict its practical applicability. I recommend a focused revision addressing the following:

- Inclusion of prediction intervals for design estimates
- Clarification or expansion of the uncertainty framework
- Consideration or discussion of alternative statistical approaches

Without adding excess length, we have addressed each of these suggestions as explained above.

**Figures**

Some supplementary figures (e.g., S3–S7) remain hard to interpret due to dense overlays or unclear legends. Improving their clarity would enhance the presentation and facilitate the interpretation of the results.

All figures have been revised, noting these comments and taking into account colour-blindness considerations.

The figures lack visual consistency in terms of font style, axis scale, and label size. A uniform formatting standard across all figures would improve readability and presentation quality.

**Methodological Choices**

The continued reliance on linear regression—despite suggestions to consider more flexible or robust methods such as quantile regression—is a missed opportunity. Given the goal of estimating extremes and the relatively low R² values reported, this limitation should at least be acknowledged and addressed as a direction for future work.

The previous version of the paper had noted alternatives, including spatiotemporal modelling for example. The new discussion section considers opportunities for quantile regression, and notes how the limitations of this data set may limit the potential of using other methods.

**Data Limitations**

The manuscript rightly notes that many sites have short records (7–10 years), and data are pooled across sources and decades. However, it does not assess how these factors affect flood frequency estimation or prediction accuracy. Similarly, the use of annual maxima overlooks possible seasonal structures or secondary peaks, which may lead to underestimation in some basins. Please reflect on these points in the manuscript text.

A new paragraph has been added at the start of section 6.4 to note how further analysis of streamflow may be useful to enhance understanding.

Overall, we believe this revised version is substantially improved as a result of the reviewers' input and the refinements we have made.

Finally, we affirm that this manuscript is original, has not been published previously, and is not under consideration for publication elsewhere. We have no conflicts of interest to disclose. All authors have read and approved the manuscript, and agree to its re-submission.

Thank you for your consideration.

Yours sincerely,

Pamela Tolentino
On behalf of the Catchment Project research team